# Early Palaeozoic ocean anoxia and global warming driven by the evolution of shallow burrowing

Sebastiaan van de Velde [1,2], Benjamin J.W. Mills [3], Filip J.R. Meysman[2,4], Timothy M. Lenton[5] & Simon W. Poulton [3]

The evolution of burrowing animals forms a defining event in the history of the Earth. It has been hypothesised that the expansion of seafloor burrowing during the Palaeozoic altered the biogeochemistry of the oceans and atmosphere. However, whilst potential impacts of bioturbation on the individual phosphorus, oxygen and sulphur cycles have been considered, combined effects have not been investigated, leading to major uncertainty over the timing and magnitude of the Earth system response to the evolution of bioturbation. Here we integrate the evolution of bioturbation into the COPSE model of global biogeochemical cycling, and compare quantitative model predictions to multiple geochemical proxies. Our results suggest that the advent of shallow burrowing in the early Cambrian contributed to a global low-oxygen state, which prevailed for ~100 million years. This impact of bioturbation on global biogeochemistry likely affected animal evolution through expanded ocean anoxia, high atmospheric $CO_2$ levels and global warming.

[1] Department of Chemistry, Analytical, Environmental and Geo-Chemistry, Vrije Universiteit Brussel, Pleinlaan 2, 1050 Brussel, Belgium. [2] Ecosystem Management Research Group, Department of Biology, Universiteit Antwerpen, Universiteitsplein 1, 2610 Wilrijk, Belgium. [3] School of Earth and Environment, University of Leeds, Leeds LS2 9JT, UK. [4] Department of Biotechnology, Delft University of Technology, Van der Maasweg 9, 2629 HZ Delft, The Netherlands. [5] Earth System Science Group, College of Life and Environmental Sciences, University of Exeter, EX4 4QE Exeter, UK. Correspondence and requests for materials should be addressed to S.V.D.Velde (email: sebastiaan.van.de.velde@vub.be)

Most of the oxygenated seafloor within the present-day ocean is inhabited by benthic animals that influence sediment biogeochemistry due to movement, ingestion of food particles and the construction of burrows[1–3]. This faunal reworking of the seafloor is termed 'bioturbation' and, following its definition for present-day sediments, the process has two separate effects[4]: the upwards and downwards transport of solid phase minerals and particles (bio-mixing), and the exchange of pore water solutes with the overlying water (bio-irrigation).

Burrowing fauna appeared in the Cambrian[5], marking a transition from the largely undisturbed microbial mat coverings of the Neoproterozoic to a colonised and reworked seafloor in the Cambrian[2,3,6]. Whilst it is still unclear when bioturbation reached modern-day mixing depths, recent work has indicated that this likely occurred gradually throughout the Palaeozoic[3,7]. The mixed layer depth through the early Palaeozoic was relatively shallow, around 1–3 cm on average[7]. Although these burrowing depths remain well below the tens of centimetres that are frequently encountered today[8], this was a fundamental change from the Precambrian, when burrowing and sediment reworking was minimal[5,6].

Bioturbation influences biogeochemical cycling in marine sediments in a number of ways. Foremost, bioturbation induces 'redox oscillations' in the upper sediment horizons, as bio-mixing transports deeper anoxic sediment back to the oxic zone near the sediment-water interface, thus re-exposing previously buried organic matter to oxygen. This continuous cycling between oxic and anoxic sediment horizons leads to a more complete breakdown of organic matter, thus reducing the rate of burial of organic carbon ($C_{org}$) on the seafloor[9–11]. Similarly, bio-irrigation introduces oxygen-rich water into deeper anoxic zones via burrow flushing, thus stimulating aerobic respiration[12] and increasing the net carbon mineralisation rate[13]. Bioturbation also increases the oxygen exposure of other redox-sensitive species in the sediment, such as pyrite ($FeS_2$), thus enhancing oxidative reaction pathways of sulphur and iron[14–16]. The resulting reduction in the burial rate of $FeS_2$ has led to the suggestion that the advent of bioturbation caused an increase in oceanic sulphate concentrations in the Phanerozoic relative to the Precambrian[16,17].

In addition to the carbon and sulphur cycles, bioturbation also influences the marine phosphorus cycle, with important consequences for oceanic productivity[3]. The enhanced oxygen influx from bio-irrigation substantially increases the oxic volume of sediment[18], and hence expands the redox niche under which microbial polyphosphate synthesis occurs, while bio-mixing also diverts labile organic phosphorus away from aerobic sediment layers, thus leading to enhanced preservation of organic phosphorus ($P_{org}$)[19]. Therefore, in bioturbated sediments underlying oxic waters, $C_{org}:P_{org}$ ratios are typically in the range of 30–115, which is lower than for laminated sediments underlying anoxic waters, where $C_{org}:P_{org}$ ratios often amount to 200–700 in modern settings[20], and up to 3500 in the geological record[21]. A recent model analysis of this phosphorus feedback suggests that the rise of bioturbation in the Cambrian period (541–485 million years ago, Ma) may have driven atmosphere and ocean de-oxygenation by increasing phosphorus preservation in sediments[19,22].

Given the key role of the seafloor in constraining Earth's geochemistry on geological timescales[23,24], and the known impacts of bioturbation on sedimentary cycling, this has provoked the idea that the appearance of burrowing animals may have substantially changed the global geochemical cycles of carbon, sulphur, phosphorus and oxygen[3,16,17,22]. Although the qualitative nature of the feedback of bioturbation on global biogeochemical cycling is recognised[3,5], the timing and magnitude of this feedback remain highly uncertain, which significantly limits our ability to reconstruct the chemical and biological response of the biosphere to early animal evolution. Conventionally, it has been assumed that the biogeochemical effect of bioturbation is more important when bioturbation parameters (the mixed layer depth and mixing intensity) reach near-modern values. As a consequence, it has been suggested that the major environmental impact of bioturbation was delayed until the Devonian, when major increases in burrowing occurred, more than 120 million years after the Cambrian 'explosion' of animal life[7]. Recently, however, both experimental[25] and modelling studies[16] have shown that biogeochemical processes respond non-linearly to bioturbation, and furthermore, that large biogeochemical impacts can occur even at shallow mixed layer depths and with low mixing intensities. Indeed, some of the most drastic changes in sediment biogeochemistry occur at low bioturbation levels[16]. This suggests that shallow bioturbation may have exerted a strong biogeochemical impact during the early Palaeozoic.

Here we present an evaluation of the global Earth system response to the rise of bioturbation, using a global biogeochemical model (COPSE)[26,27] that simulates the coupled cycling of carbon, oxygen, phosphorus and sulphur. Based on the known impact of bioturbation in modern sediments, we implement a new parameterisation for bioturbation in the COPSE model that affects the cycles of carbon, oxygen and phosphorus. We then ground-truth our model outputs in relation to multiple types of geochemical data from the rock record. This allows us to reconcile contrasting views regarding the effects of bioturbation on global elemental cycles during the Cambrian, thus providing new insight into a significant potential driver of environmental change at a pivotal juncture in Earth history.

## Results

**Geochemical evolution of the early Palaeozoic Earth system.** Geochemical data covering the late Neoproterozoic and early Phanerozoic (560–420 Ma) are summarised in Fig. 1. In general, there is evidence for considerable heterogeneity in the evolution of ocean redox chemistry from the late Neoproterozoic through to the mid-Cambrian, as might be expected in a pervasively low oxygen world. However, much of this apparent heterogeneity may be due to difficulties in adequately sampling sediments from a range of water depths in individual studies. For example, the existence of an anoxic oxygen minimum zone (OMZ) along productive continental margin settings has been advocated for the early Cambrian[28]. Samples from within the OMZ would give a very different redox signature compared to samples from oxic shallower and deeper waters. Nevertheless, an overall progression in ocean oxygenation across this period is now emerging, aided by the application of redox indicators that provide a more global indication of ocean redox chemistry[29–31].

There is robust evidence for widespread deep ocean oxygenation in the late Neoproterozoic[30,32–35]. Building upon this, selenium isotope evidence[30] also suggests progressive oxygenation through the Neoproterozoic. However, multiple lines of evidence demonstrate a short-lived return to widespread ocean anoxia at the Precambrian-Cambrian boundary (Fig. 1a and Supplementary Fig. 1)[33,36–38]. This anoxic episode was likely too short-lived to be captured by our modelling approach (see below), but evidence from sedimentary Mo concentrations and isotopes, U isotopes, and rare earth element concentrations suggests that the global ocean then became progressively oxygenated through the early Cambrian up until the height of the Cambrian explosion at ~520 Ma[29,31,34]. These same redox proxies demonstrate a subsequent return to more widespread anoxia after ~520 Ma

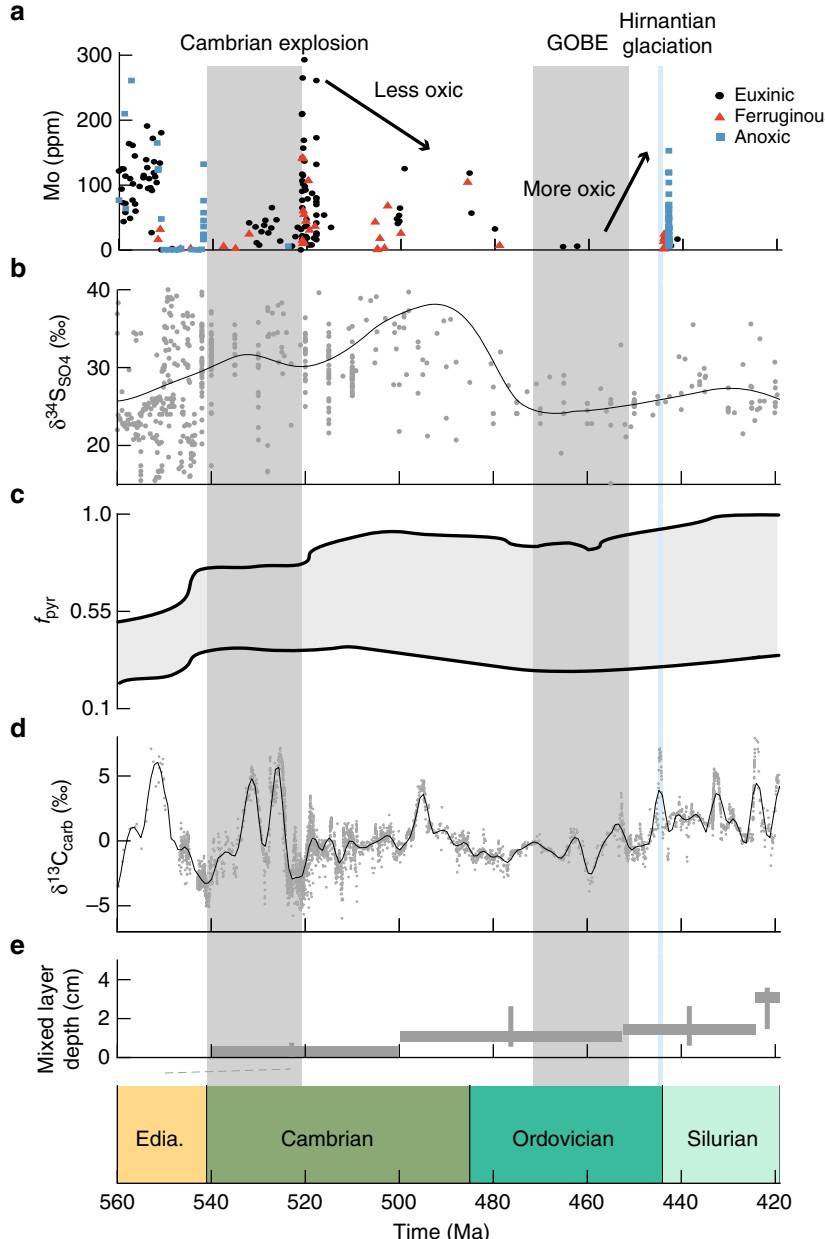

**Fig. 1** Compilation of geochemical data from the Neoproterozoic into the Palaeozoic. **a** Mo abundances, as compiled by ref. [29]. Higher oxygen levels lead to higher abundances of Mo. Note that other proxies providing support for the ocean oxygenation state outlined here are compiled in Supplementary Fig. 1. **b** Sulphate-S isotopes ($\delta^{34}S_{SO4}$) as compiled by ref. [69]. $\delta^{34}S_{SO4}$ increases through the Cambrian, with a return to lighter values at the GOBE. **c** Pyrite fraction of sulphur burial ($f_{pyr}$ = pyrite burial/(pyrite burial + gypsum burial)), full range of estimates, as presented in ref. [44]. **d** Carbonate-C isotopes ($\delta^{13}C_{carb}$) as compiled by ref. [70]. Higher values indicate a higher rate of organic carbon burial. **e** Mixed layer depth, reproduced from ref. [7]. Black lines on panels **b** and **d** are local regression (LOESS) fits. Grey shaded areas indicate the Cambrian explosion (540–521 Ma) and the Great Ordovician Biodiversification Event (GOBE; 470–450 Ma). Blue shaded line indicates the Hirnantian glaciation

(for simplicity we demonstrate this with Mo concentration data in Fig. 1, but other proxies are compiled in Supplementary Fig. 1), which is also consistent with evidence from S isotopes, Fe speciation, and trace metal concentrations indicating widespread euxinia in the later Cambrian ocean[39].

Evidence from sulphur isotope systematics suggest this widespread euxinia continued during the early and mid-Ordovician[40]. Other geochemical records through the Ordovician are relatively sparse, although ocean redox conditions appear to have been subject to temporal variability, which is consistent with an overall low-oxygen world[41]. However, by the end of the Ordovician and during the Silurian (<460 Ma), oxygen started to increase[27],

coincident with the earliest instance of fossilised charcoal, indicating near-modern levels of atmospheric $O_2$[42]. This rise in oxygen has been attributed to the evolution of land plants, which culminated in the Devonian and Carboniferous with the development of roots and vasculature[27,43].

The sedimentary record of oceanic sulphate $\delta^{34}S$ shows significant variability (Fig. 1b), but the general trend suggests an increase in $\delta^{34}S_{SO4}$ from the Ediacaran (25‰) to the mid Cambrian (39‰), followed by a recovery to pre-Palaeozoic values of 25‰, just before the Great Ordovician Biodiversification Event (GOBE). Increases in $\delta^{34}S_{SO4}$ are often linked to increased rates of pyrite burial, but this may not necessarily be the case during

the Early Palaeozoic[44] (note the very large range in $f_{pyr}$; the fraction of ocean sulphate that is buried as pyrite; Fig. 1c). A long-term perturbation during the Cambrian and Ordovician is apparent in the sedimentary carbonate $\delta^{13}C$ record. While this record reveals fluctuations in the carbon cycle on short timescales of a few million years, on a longer time scale the $\delta^{13}C_{carb}$ record shows generally lower values from the middle Cambrian to the late Ordovician (~0‰) as compared to the late Ediacaran and early Cambrian (~4‰), with a return to higher values in the Silurian (~2‰). The shift in $\delta^{13}C_{carb}$ at around 521 Ma is consistent with a reduction in net organic carbon burial, and a resultant decrease in the production rate of $O_2$. The recovery to more oxic conditions in the Silurian was likely coupled to increased organic carbon burial, as indicated by the more positive $\delta^{13}C_{carb}$ values (Fig. 1d).

The analysis of sediment fabric disturbance suggests early animals only reworked the seafloor superficially, with shallow burrowing appearing during the Cambrian and continuing into the Ordovician, whereas only towards the end of the Silurian do burrow systems become deeper (mixed layer depth shown in Fig. 1e)[7]. Various scalings of the biogeochemical response to bioturbation have been shown to be consistent with subsets of available geochemical proxies: a rapid and non-linear response at low bioturbation intensities may have increased phosphate retention in the sediment when the first shallow-burrowing animals appeared in the Cambrian[22], leading to a decrease in the oxygen production source from organic carbon burial, and driving a return to anoxic ocean conditions after 520 Ma (Fig. 1a). In contrast, a more protracted response has been shown to be consistent with low sulphate concentrations throughout the Palaeozoic[7]. However, the proposed scenarios for the evolution of bioturbation have not been evaluated using multiple geochemical proxies or model simulations combining feedbacks between the carbon, oxygen, phosphorus and sulphur cycles.

**The COPSE model.** To obtain more robust constraints on the timing and environmental consequences of the rise of bioturbation, we modified the COPSE model[26,27] (Carbon Oxygen Phosphorus Sulphur Evolution), which is a synthesis of the 'Redfield' model[45] and the GEOCARB model[46]. The COPSE model simulates the coupled evolution of the major biogeochemical cycles over the Phanerozoic by describing burial, weathering and degassing processes, which transport chemical species between the atmosphere, oceans and sediments over geological timescales (Fig. 2). COPSE aims to capture trends in biogeochemical cycling over the timescale

of 10–100 s of millions of years, but not shorter-term fluctuations on timescales of ~100,000 years. The model produces estimates for the global abundance of oxygen, carbon dioxide, phosphate and sulphate, alongside records of whole-ocean $\delta^{13}C_{carb}$ and $\delta^{34}S_{SO4}$, which can be used to test hypotheses by comparison to data. For an overview of the biotic and tectonic controls covered by the current COPSE model, see ref. [47]. The evolution of bioturbation and its feedback on global biogeochemistry has not been explicitly considered in COPSE: previous model versions have implicitly assumed that bioturbation is always active and that the bioturbation intensity remains independent of the oceanic oxygenation state.

By using COPSE we employ a forward modelling approach, which enables a comparison of model predictions of $\delta^{13}C_{carb}$ and $\delta^{34}S_{SO4}$ trends to the independent geological record. This contrasts with inverse modelling, where geological records are used as a model forcing, leaving no potential for quantitative testing of the model results. As with all models, comparison with the geological record requires some assumptions. Foremost, the model predicts $\delta^{13}C_{carb}$ and $\delta^{34}S_{SO4}$ trends that are representative of the global marine dissolved inorganic carbon and sulphate reservoirs, reflecting the globally averaged operation of the long-term geochemical cycles (in essence changes in organic carbon and pyrite burial). However, changes in the geological isotope record are not solely dependent on changes in the global biogeochemical cycling of carbon and sulphur, but also incorporate possible effects of diagenesis, or evolutionary changes to the fractionation factors associated with photosynthesis and microbial sulphate reduction. Moreover, it is possible that some data represent regional signals rather than global trends. Nevertheless a quantitative comparison of our model predictions to the geological isotope record provides a useful test of the assumptions underlying the COPSE model.

For the late Ediacaran to mid Ordovician, the 'baseline' COPSE model (i.e., the model version as presented in ref. [27]) generates stable conditions with high atmospheric $CO_2$ (~15× Present Atmospheric Level (PAL)), a high degree of ocean anoxia (0.8, or 80% of the ocean surface resides under anoxic water) and $\delta^{13}C_{carb}$ around 0‰ (Fig. 3). Broadly, these results are driven by the absence of terrestrial productivity, and a suppression of silicate mineral weathering before land plant evolution (i.e., suppressed burial of both organic and inorganic carbon). The predictions of the baseline model reveal discrepancies with the available geochemical data, which suggest more dynamic ocean redox conditions in the early Palaeozoic and higher $\delta^{13}C_{carb}$ values in the late Ediacaran and early Cambrian (Fig. 1).

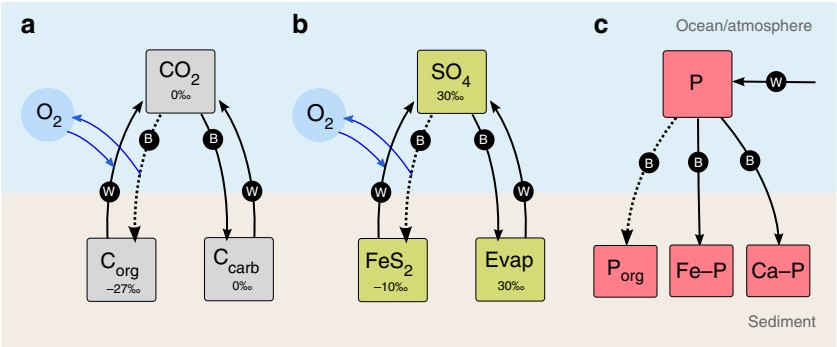

**Fig. 2** Diagram of key processes in the COPSE model. **a** Carbon cycle. Hydrospheric $CO_2$ is transferred to sediments as organic C or carbonate by burial (B). Sedimentary C is returned to the ocean/atmosphere via weathering and metamorphism (W). Buried organic C is isotopically lighter than the carbon it is derived from. Burial of reduced organic carbon results in a net source of $O_2$, whereas oxidative weathering of sedimentary organic carbon consumes $O_2$. **b** Sulphur cycle. Burial of reduced pyrite is a net source of $O_2$, whereas oxidative weathering of sedimentary pyrite consumes $O_2$. **c** Oceanic phosphorus (P) cycle. Dissolved, bio-available P is delivered to the ocean by chemical weathering via rivers, and is buried either as organic phosphorus, or with iron or calcium minerals. Dashed lines show burial processes that are influenced by bioturbation (but are not considered so in the baseline model)

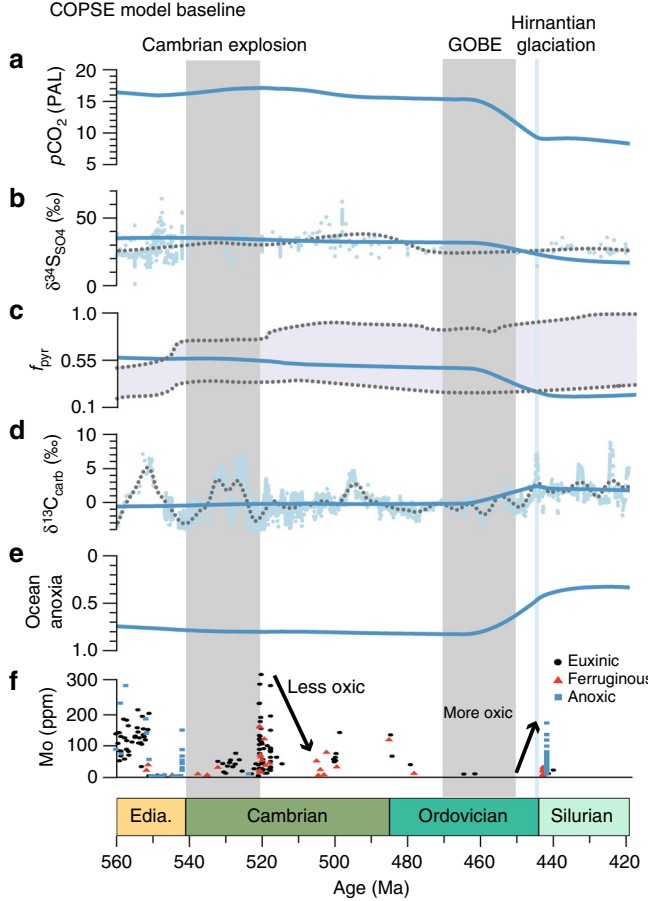

**Fig. 3** COPSE baseline model simulation. Simulation as presented in ref. [27]. **a** Atmospheric $CO_2$. **b** Average $\delta^{34}S_{SO4}$ of seawater. **c** Pyrite fraction of sulphur burial. **d** Average $\delta^{13}C_{carb}$ of seawater. **e** Degree of ocean anoxia (1 = completely anoxic, 0 = completely oxic). **f** Summary of the evolution of sedimentary Mo concentrations over time. Model outcomes (in blue) are compared to $\delta^{13}C_{carb}$ and $\delta^{13}S_{SO4}$ data and the sedimentary Mo concentrations, which is reflective of the extent of ocean oxygenation and is supported by multiple independent proxies (see Supplementary Note 1). Dotted lines in panels **b** and **d** represents a local regression (LOESS) fit to the data. Grey shaded areas indicate the Cambrian explosion (540–521 Ma) and the Great Ordovician Biodiversification Event (GOBE; 470–450 Ma). Blue shaded line indicates the Hirnantian glaciation

**The effect of bioturbation on sedimentary elemental cycling.** Here, we update the COPSE model parameterisation for the burial of organic carbon, pyrite sulphur and phosphorus to include a response to bioturbation. We introduce a bioturbation parameter, $f_{biot}$, which provides a single measure of the biogeochemical impact of bioturbation on sedimentary cycling (0 represents no bioturbation effect, 1 represents maximum bioturbation impact). This $f_{biot}$ parameter must be expressed as a function of the strength of bioturbation, which is traditionally represented by the bio-mixing depth ($L_b$) and the bio-mixing intensity ($D_b$) parameters, that feature in early diagenetic models[48]. As $D_b$ and $L_b$ are essentially linked ($L_b \sim \sqrt{D_b}$, see ref. [49] for a theoretical justification), the parameter $f_{biot}$ may be expressed as some function of either one of these parameters (Fig. 4). Currently, there is a lack of data to constrain the exact nature of the relation between $f_{biot}$ and bioturbation intensity.

To examine the link between the emergence of bioturbation and global biogeochemistry, we describe three different formulations for the evolution of $f_{biot}$ over time that are designed to

represent the envelope of possibilities. In 'Scenario 1', the biogeochemical impact of bioturbation is assumed to be weak during the Cambrian and Ordovician, and becomes important when deeper and more intensively burrowing organisms evolve in the late Silurian and Devonian, increasing the mixing depth and creating larger burrow networks that are intensely flushed with overlying water (solid line in Fig. 4b). Throughout the Cambrian and Ordovician, the mixed layer depth may have been at least 5 times lower than the observed depths in the late Silurian and Devonian (Fig. 1d and ref. [7]), and therefore we choose $f_{biot} = 0.2$ (thereby assuming a linear relationship between the effects of bioturbation and the mixed layer depth). Mixed layer depths likely increased to higher levels in the late Silurian and the Devonian[7]. Therefore, we allow $f_{biot}$ to increase to 0.5 by the end of the Silurian, which implies that the biogeochemical impact of bioturbation only peaked after the Silurian (as suggested by ref. [7]).

In Scenarios 2 and 3, we suggest that low depths of bioturbation mediate a disproportionally large response in sediment geochemistry. Both experimental[25] and theoretical[16] studies have proposed that geochemical state variables and rates (e.g., elemental cycling rates, partitioning of redox acceptors, stimulation of organic carbon breakdown) respond in a highly non-linear way to increasing levels of bioturbation, with 80–90% of the maximum response attained at low bioturbation intensities ($D_b < 1$ cm$^2$ yr$^{-1}$ and $L_b < 3$ cm; see ref.[16]). For example, diagenetic modelling shows that the cycling rate of sulphur (the number of times a sulphur atom entering the sediment column is cycled between its oxidised and reduced states before it is eventually buried) rapidly increases from 5 to 10 when $D_b$ increases from 0 to 1 cm$^2$ yr$^{-1}$, to stabilise at 11 when $D_b$ values > 1 cm$^2$ yr$^{-1}$ (Fig. 4a). This increased redox cycling of sulphur potentially inhibits the rate at which reduced sulphur compounds are buried (by stimulating re-oxidation)[14,17]. Additionally, experimental work has shown that meiofauna (micron-scale animals that burrow ~1 cm) stimulate organic carbon breakdown as much as large animals (that burrow >10 cm depth)[25].

During the Cambrian, the mixed layer depth increased from 0 to <0.5 cm [7], which may correlate with a significant biogeochemical response. In Scenario 3, the advent of shallow mixing thus invokes a large biogeochemical effect, and so the bioturbation impact ($f_{biot}$) increases exponentially from 0 to 1 during the Cambrian Explosion, and remains constant afterwards (dash-dotted line in Fig. 4b). We also define an intermediate situation (Scenario 2), in which we assume that shallow bioturbation has a large impact, but that the areal extent of bioturbation (and accordingly $f_{biot}$) gradually increases throughout the early Palaeozoic (dashed line in Fig. 4b). Furthermore, in all scenarios, two distinct options for the response of bioturbation towards anoxia have been tested; (i) no anoxia limitation, and (ii) anoxia limitation that scales with the fraction of the ocean that is anoxic (solid vs. dashed lines in model outputs, see Methods for more information).

The effect of bioturbation on the elemental cycling of C, P and S in marine sediments is summarised in Fig. 4c. Overall, the main effects of bioturbation are driven by the increase in oxygen exposure in the anoxic part of the sediment. In a sediment without bioturbation (e.g., the Ediacaran seafloor), organic matter is broken down less efficiently[11,50,51] and sulphur is more efficiently sequestered as pyrite, leading to high burial rates of organic carbon and pyrite. With the introduction of burrowing fauna, organic matter mineralisation is enhanced and pyrite is more efficiently re-oxidised[14], and so the burial of carbon and sulphur is reduced[3,17]. At the same time, bioturbation leads to an increase in polyphosphate sequestering, which then leads to an increase in organic phosphorus burial[19]. We implemented

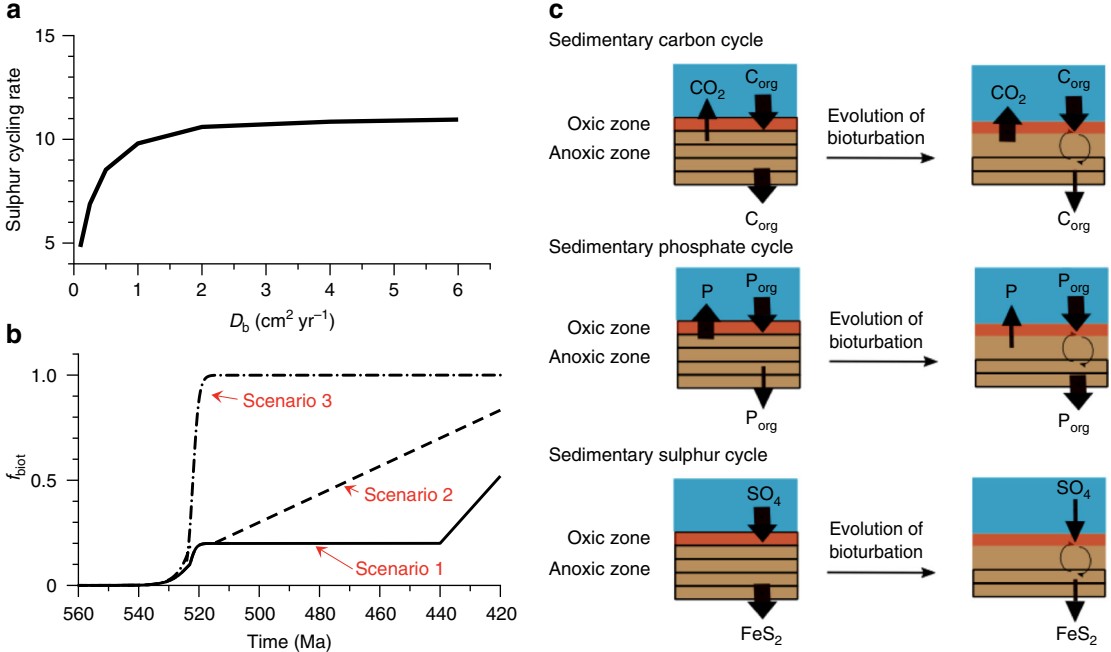

Fig. 4 The effect of bioturbation on sediment geochemistry. **a** The effect of bioturbation intensity ($D_b$) on the sulphur cycling rate. Model results reproduced from ref. [16]. **b** Solid line (Scenario 1): the bioturbation impact on geochemistry ($f_{biot}$) is linearly correlated with the depth and intensity of burrowing. Dashed line (Scenario 2): the bioturbation impact is maximal with shallow burrowing, but the areal expansion of bioturbation increases gradually throughout the early Palaeozoic. Dash-dotted line (Scenario 3): the bioturbation impact on geochemistry ($f_{biot}$) is already at full strength by the end of the Cambrian explosion. **c** Simplified conceptual model of the effect of bioturbation on the sedimentary cycles of carbon, phosphorus and sulphur. Arrow sizes denote relative changes in flux sizes

these relations between bioturbation and geochemical C, S and P cycling in COPSE by adapting the model equations for marine organic carbon burial, marine pyrite sulphur burial and marine organic phosphorus burial, and introduce new parameters that describe pre-bioturbation values for organic carbon, pyrite sulphur and organic phosphorus burial (see Methods).

Overall, this model description allows us to test the working hypotheses; (i) Scenario 1: the effect on global biogeochemical cycles scales with bioturbation depth and intensity, and increased only markedly in the Silurian–Devonian, (ii) Scenario 2: the effect on global biogeochemical cycles increased gradually throughout the Palaeozoic, and (iii) Scenario 3: the effect on global biogeochemical cycles was rapid and occurred in the Early Cambrian. We tested the validity of these hypotheses by comparing the quantitative model output to the geological record.

**New model results**. Under a protracted response to bioturbation (Scenario 1; Fig. 5a–g), $\delta^{13}C_{carb}$ values slightly decrease with the early development of bioturbation at the start of the Cambrian, and remain constant throughout the Cambrian and Ordovician. After rising in the late Ordovician, $\delta^{13}C_{carb}$ drops gradually during the Silurian, in response to decreased primary production and decreased carbon burial due to bioturbation (Fig. 5d). In the model simulation of ocean anoxia, the limited effects of bioturbation during the early Palaeozoic imply a high rate of organic carbon burial and therefore higher concentrations of atmospheric and oceanic oxygen, and a limited prevalence of anoxia (Fig. 5e). Both of these predictions are to some extent at odds with the geochemical record, which shows evidence for significant ocean anoxia after 520 Ma [31] until the middle Ordovician (470–460 Ma) [40], and does not support high $\delta^{13}C_{carb}$ (~2‰) throughout this interval (Fig. 1).

With a gradually increasing areal extent of bioturbation across the Palaeozoic (Scenario 2; Fig. 5h–n), organic matter burial gradually decreases, while sedimentary phosphate retention gradually increases, both leading to an increase in ocean anoxia (Fig. 5k, l). While the associated protracted decrease in $\delta^{13}C_{carb}$ values is not entirely at odds with the geochemical record, the gradual increase of ocean anoxia across the Palaeozoic does not fully agree with geochemical proxy evidence for an increase in ocean anoxia in the early Cambrian [31]. Furthermore, the absence of a transient increase in $\delta^{34}S_{SO4}$ also disagrees with the geochemical record (Fig. 5i).

When the model simulations incorporate a strong biogeochemical response to shallow bioturbation in the early Cambrian (Scenario 3; Fig. 5o–u), the emergence of bioturbation results in significantly enhanced oxidation of marine organic carbon and pyrite, as well as benthic phosphate retention, which limits oceanic primary production. The initial decrease in marine organic carbon burial at 520 Ma (marked by the drop in $\delta^{13}C_{carb}$ from 2‰ to 0‰; Fig. 5r) is accompanied by an increase in ocean anoxia (Fig. 5s), which then provides a negative feedback and limits bioturbation. The resulting reduction in marine organic carbon burial results in increased atmospheric $pCO_2$ (by >1000 ppm; Fig. 5o), and suggests that the evolution of burrowing organisms in the ocean could have triggered significant climate warming, consistent with the 'greenhouse' climate invoked for the Cambrian and early Ordovician, as indicated by elevated sea levels [52] and oxygen isotope systematics [53]. We note that the direct effect of the organic carbon cycle on $CO_2$ levels is not as widely discussed in the literature as the changes resulting from silicate weathering, but that it is an important part of the coupled C and O cycles [54].

The increase in anoxia induced by bioturbation leads to an increase in $\delta^{34}S_{SO4}$ (~30‰; Fig. 5p). We do not find, as previously suggested [17], a rise in marine sulphate concentrations

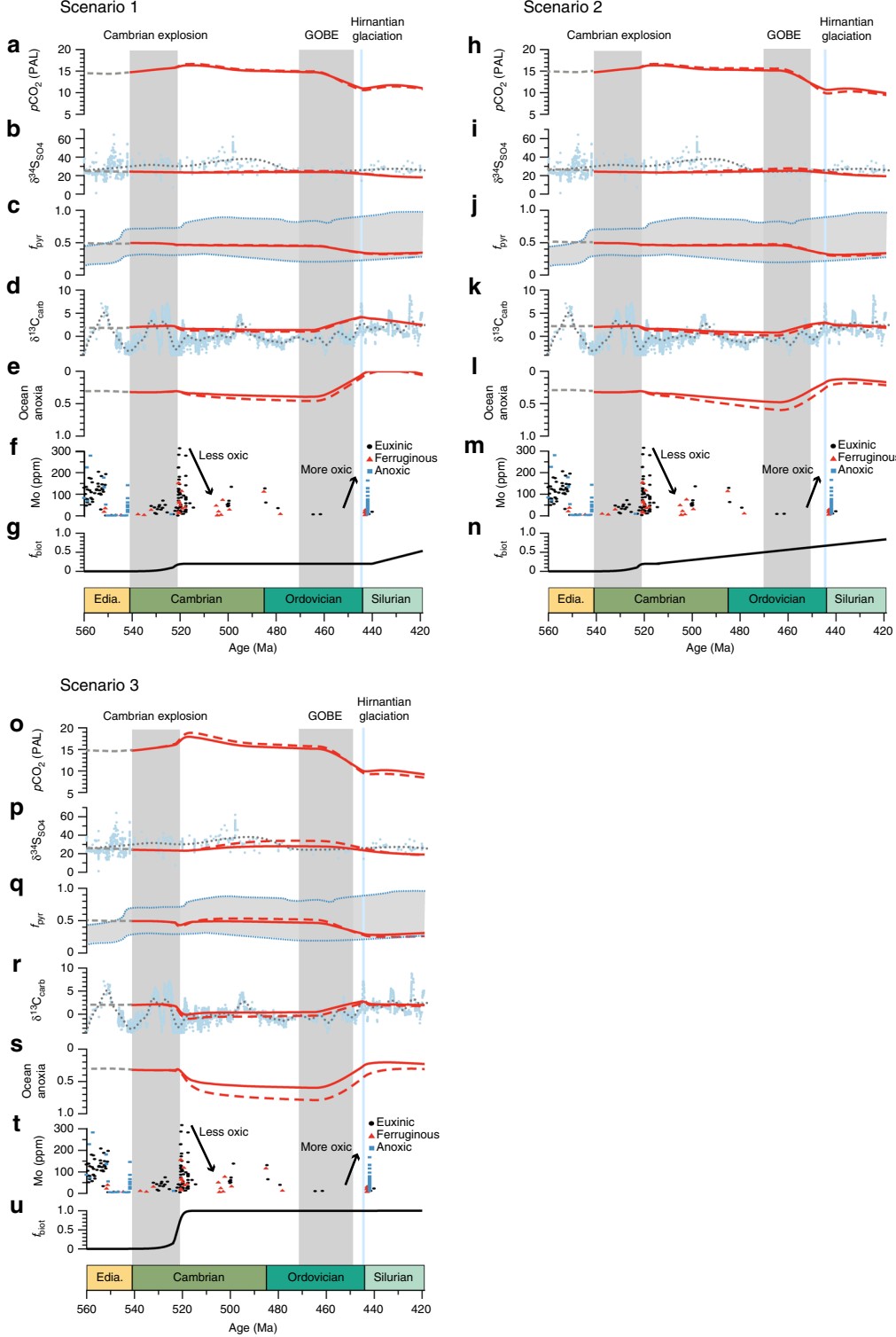

**Fig. 5** COPSE model with the addition of the evolution of bioturbation. Scenario 1 shows the effect of the sedimentary response that scales linearly with bioturbation intensity. Scenarios 2 and 3 assume that the effects of bioturbation on sediment geochemistry occur non-linearly (strong response for low levels of bioturbation), where Scenario 2 follows a gradual increase of the areal extent of bioturbation and Scenario 3 shows the maximum effect at the Ediacaran-Cambrian boundary (see panels **g**, **n**, **u**). **a**, **h**, **o** Atmospheric $CO_2$. **b**, **i**, **p** Average sulphate $\delta^{34}S$ of seawater. **c**, **j**, **q** Pyrite fraction of sulphur burial. **d**, **k**, **r** Average $\delta^{13}C$ of carbonate. **e**, **l**, **s** Degree of ocean anoxia. Model outcomes (in red) are fitted to the $\delta^{13}C_{carb}$ and $\delta^{13}S_{SO4}$ proxies (grey dotted line represents a LOESS fit), predictions for the relative importance of pyrite for the total sulphur burial rate (blue dotted lines represent the range of model results presented in ref. [44]) and compared to a summary of the evolution of sedimentary Mo concentrations over time (**f**, **m**, **t**). Solid lines represent the model outcomes with anoxia feedback, dashed lines represent the model outcomes without anoxia feedback. Grey shaded areas indicate the Cambrian explosion (540–521 Ma) and the Great Ordovician Biodiversification Event (GOBE; 470–450 Ma). Blue shaded line indicates the Hirnantian glaciation

coincident with the evolution of bioturbation (Supplementary Fig. 2), because of our implementation of gypsum (CaSO$_4$) burial. In COPSE, gypsum burial scales with the oceanic sulphate concentration, while gypsum burial was previously considered important only at higher sulphate concentrations[17]. Given the uncertainty over gypsum formation, it is difficult to make any strong conclusions regarding this effect. In general, the effect of the evolution of bioturbation on ocean sulphate concentration remains uncertain. Eventually, the effects of the evolution of bioturbation are largely reversed by the rise of early land plants in the late Ordovician, which increases net carbon burial and phosphate supply, as well as drawing down atmospheric CO$_2$. Assumed high Palaeozoic erosion rates contribute to low $\delta^{13}C_{carb}$ in the model runs[54], but it is unlikely that the rapid drop in $\delta^{13}C_{carb}$ at the end of the Cambrian explosion and the subsequent rise at the end of the Ordovician can be driven solely by rapid erosion and sedimentation, which continues into the Devonian. Overall, Scenario 3 brings the COPSE model predictions reasonably close to the available data from the geological record.

Caution is of course required, as bioturbation is likely not the only driver for variation in the isotope data, especially for variations on timescales <1 million years, which the model cannot capture. However, these limitations do not invalidate the hypothesis testing here, which shows that Scenario 3 (large geochemical effects due to shallow burrowing) can reproduce the broad changes seen in $\delta^{13}C_{carb}$, $\delta^{34}S_{SO4}$ and ocean anoxia, whilst Scenarios 1 and 2 (more protracted global biogeochemical responses) produce discrepant predictions for $\delta^{13}C_{carb}$ or $\delta^{34}S_{SO4}$. Reversing these conclusions would require a very specific, and unlikely, set of additional forcing factors (see Supplementary Note 2 for further discussion). Nevertheless, many additional mechanisms were likely at work over the studied time interval. For example, nutrient stress and limited primary production were undoubtedly a factor in explaining the low-oxygen conditions in the early Palaeozoic. The COPSE model attempts to take into account such mechanisms: for example it includes process-based ocean nutrient cycles, including a representation of phosphorus removal with organic matter, calcium, and iron species and how these sinks may have varied with time[26]. However, as with all box models, these processes are greatly simplified, and apparent mismatches with geochemical data suggest that there are still processes missing or poorly represented (e.g., the P cycle). The COPSE model predictions are improved by our additional representation of bioturbation, suggesting that it was a major driver of biogeochemical cycling during the Palaeozoic.

**Synthesis**. By recognising that moderate levels of shallow bioturbation have a large impact on sediment geochemistry (Fig. 4a), our model results are able to broadly reconcile the bioturbation record (Fig. 1e) in relation to various proxies in the geochemical record (Fig. 5). We propose a significant bioturbation-driven step change in environmental conditions and geochemical cycling in the early Cambrian, well before benthic fauna reached their full capacity in terms of sediment reworking. This appears to have resulted in a ~100 Myr period of prevalent ocean anoxia and greenhouse climate conditions that is consistent with the available geological evidence. This transition period between the Cambrian explosion and the Great Ordovician Biodiversification Event (GOBE) was also marked by sizable fluctuations in $\delta^{13}C_{carb}$ values and variable ocean redox conditions (Fig. 1) on shorter timescales, alongside repetitive extinction and recovery events that sustained a radiation plateau[55] which was eventually followed by the GOBE[56].

There is a strong correlation between ocean anoxia, positive $\delta^{13}C_{carb}$ excursions and extinction events[57], and it is possible that the interactions between burrowing macrofauna, biogeochemical cycling and ocean anoxia may have contributed to these patterns. The expansion of oceanic anoxia ultimately limits the habitable area of seafloor, and if anoxia resulted in an extinction event, the associated collapse of bioturbation would be expected to drive a positive carbon isotope excursion. In this way, the evolution of burrowing may have contributed to the diversification of Metazoa[58], since the dynamic redox conditions that appear to be a feature of the Ediacaran and Cambrian (and potentially modified by the evolution of shallow bioturbation) would have led to enhanced ecological stress, thus paving the way for the next 'explosion' of diversity—the Great Ordovician Biodiversification Event[59,60].

## Methods

**COPSE model equations and parameterisation**. We update the COPSE model by including a feedback of bioturbation on the burial of marine organic carbon (mocb), marine organic phosphorus (mopb) and marine pyrite sulphur (mpsb). In the baseline model, marine organic carbon burial is quadratically dependent on primary production (newP), where mocb$_0$ is an estimate for present-day marine organic carbon burial, and newP$_0$ is an estimate of present-day primary production. We introduce a new parameter (CB$_{biot}$), which represents enhanced mineralisation by bioturbation and is defined as:

$$CB = CB_{biot} + (1 - f_{biot})\left(CB_{prebiot} - CB_{biot}\right) \qquad (1)$$

where $f_{biot}$ is the bioturbated fraction of organic carbon and CB$_{prebiot}$ the mineralisation without bioturbation. The final expression is then:

$$mocb = mocb_0 \left(\frac{newP}{newP_0}\right)^2 CB \qquad (2)$$

Several experiments have shown that the presence of burrowing macrofauna reduces organic carbon burial by ~50%, which implies a significant reduction in organic C burial in bioturbated sediments[11,61], leading to CB$_{prebiot}$ = 2 × CB$_{biot}$. This number is likely not applicable to the whole ocean, although the majority of carbon burial takes place in coastal areas and continental shelves[62], which are also subject to the highest rates of bioturbation. About 80% of the global carbon burial takes place in the continental margin sediments (this takes into account the presence of relict sands that do not accumulate organic matter)[62]. Since about 5% of the near-shore seafloor is currently covered by hypoxic waters[63], this restricts benthic activity to maximum 95% of the continental margin sediments. This gives CB$_{prebiot}$ = 1.6 × CB$_{biot}$ (95% of 80% is increased by a factor 2, and the remaining 24 % is unaltered). As a more conservative estimate, we assume that only 40% of the continental shelf sediments are affected by bioturbation, which then leads to a CB$_{prebiot}$ = 1.2 × CB$_{biot}$. We tested these three values for the CB$_{biot}$ parameter (see Supplementary Note 3), and found that the higher values caused unreasonably large changes in the model due to the global and nondimensional nature of the system, which does not capture regional feedbacks. Therefore we choose the more conservative value of CB$_{prebiot}$ = 1.2 × CB$_{biot}$ for this work. It is common for processes measured at the local scale to cause overestimates when applied in global scale models, and this parameter choice does not affect our conclusions, which are based on the qualitative changes observed at different times in Earth history.

Marine organic phosphorus burial is dependent on the C:P ratio of organic carbon in bioturbated sediment (CP$_{biot}$) and non-bioturbated (or laminated) sediment (CP$_{lam}$), and the burial rate of organic carbon (mocb). Equation (3) has been introduced before[22] and represents the dependence of mopb on the bioturbated fraction of organic carbon:

$$mopb = mocb\left(\frac{f_{biot}}{CP_{biot}} + \frac{1 - f_{biot}}{CP_{lam}}\right) \qquad (3)$$

We can trial again two extreme values for CP$_{biot}$ (the C$_{org}$:P$_{org}$ ratio of bioturbated organic matter). Following the reasoning from above (today, maximum 95% of 80% of the buried organic matter experiences bioturbation) and mean values for unbioturbated and bioturbated C$_{org}$:P$_{org}$ ratios (450 and 73.5, respectively, derived from ref. [64]), we arrive at a lower bound value of CP$_{biot}$ = 160. Similarly, assuming only 40% of the 80% experiences bioturbation, we arrive at CP$_{biot}$ = 330. We tested values of 150, 250 and 350 for CP$_{biot}$ and values of 1000, 2000 and 3000 for CP$_{lam}$ (which compromises between higher palaeo data and lower modern estimates)[19,22,64]. For the model runs in the paper, we employed the conservative values CP$_{biot}$ = 250 and CP$_{lam}$ = 1000.

A final adaptation is in the marine pyrite burial (mpsb). In the baseline model, mpsb is linearly dependent on the oceanic sulphate concentration and mocb, and

**Table 1 Parameters used in the model simulations**

| Parameter | Units | Value | Ref. |
|---|---|---|---|
| $mocb_0$ | mol C yr$^{-1}$ | $4.50 \times 10^{12}$ | [26] |
| $mpsb_0$ | mol S yr$^{-1}$ | $0.53 \times 10^{12}$ | [71] |
| $[SO_4]_0$ | mol | $4.0 \times 10^{19}$ | [26] |
| $newP_0$ | mol kg$^{-1}$ yr$^{-1}$ | 225.96 | [24] |
| $CB_{prebiot}$ | — | 1.2 | This paper |
| $SC_{prebiot}$ | — | 0 | This paper |
| $CP_{biot}$ | — | 250 | This paper |
| $CP_{lam}$ | — | 2000 | This paper |

inversely dependent on oxygen concentrations. We introduce an extra parameter which represents the enhanced recycling of sulphur ($SC_{biot}$), dependent on bioturbation, which has the same form as Eq. (1):

$$SC = SC_{biot} + (1 - f_{biot})\left(SC_{prebiot} - SC_{biot}\right) \quad (4)$$

where $SC_{prebiot}$ is the rate of sulphur recycling without bioturbation. The final equation then becomes:

$$mpsb = mpsb_0 \times \frac{[SO_4]}{[SO_4]_0} \times \frac{1}{[O_2]} \times \frac{mocb}{mocb_0} SC \quad (5)$$

Bioturbation would enhance reduced sulphur reoxidation, thus stimulating sulphur recycling[3,17]. Together with decreasing organic carbon burial, this affects the $C_{org}$:$S_{pyr}$ ratio (organic carbon vs. pyrite) in the sediment[14]. Indeed, in unbioturbated sediments, pyrite formation is mainly driven by organic carbon availability, leading to fairly constant $C_{org}$:$S_{pyr}$ ratios[65]. Throughout the Phanerozoic however, the $C_{org}$:$S_{pyr}$ ratio changes, seemingly in sync with atmospheric oxygen[66]. Even though this effect on $C_{org}$:$S_{pyr}$ ratios is likely the consequence of increasing bioturbation levels throughout the Phanerozoic, the baseline model of COPSE already contains this dependency of pyrite sulphur burial in the form of the inverse dependency on oxygen concentrations[26]. Since in our model, bioturbation is also dependent on the oxygen content, we removed the direct effect of bioturbation on sulphur cycling from the more conservative model (by setting $SC_{prebiot} = 1$) as the effect of bioturbation on pyrite burial is already implemented to some degree.

The present day values of $CB_{biot}$ and $SC_{biot}$ are set to 1 by means of reference, and scaling factors are introduced to set their values to 'prebiotic' conditions with no bioturbation (Table 1). The sensitivity tests for the alternative values for $CB_{prebiot}$, $SC_{prebiot}$, $CP_{biot}$ and $CP_{lam}$ are shown in the Supplementary Note 3 (Supplementary Figs. 3, 4, 5) and the results are qualitatively robust to these choices.

**Anoxia limitation**. Oxygen is a key metabolic resource for animals[67,68] but the oxygen concentrations at which macrofauna are excluded and bioturbation is impeded are presently not well constrained (even in modern oxygen minimum zones, where dissolved $O_2$ is <9 µM, animals are found[67], but these appear to display minimal bioturbation activity). Therefore, we explored two distinct options for the response of bioturbation towards anoxia; (i) $f_{biot}$ is not dependent on the degree of oceanic anoxia and (ii) $f_{biot} = f_{biot}^{ref} (1 - anox)$, where anox is the degree of ocean Anoxia, as in the original model:

$$anox = \max\left(1 - k_{oxfrac}(RO_2)\frac{newp_0}{newp}, 0\right) \quad (6)$$

**The bioturbation parameter $f_{biot}$**. To parameterise the geochemical sediment response in our model, the bioturbation parameter has been described as a function of time, based on the three tested scenarios (see main text for more details); Scenario 1 (the effect of bioturbation on sediment geochemistry scales with the depth and intensity of burrowing), Scenario 2 (low levels of bioturbation invoke a large response in sediment geochemistry, and the areal extent of bioturbation gradually increased throughout the Palaeozoic), Scenario 3 (low levels of bioturbation invoke a disproportionally large response in sediment geochemistry).

**Data availability**. The data presented in this manuscript can be freely accessed on ResearchGate [https://www.researchgate.net/publication/325361821_Data_from_van_de_Velde_et_al_Nature_Communications_2018].

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

## Acknowledgements
Research was financially supported by the European Research Council under the European Union's Seventh Framework Programme (FP/2007–2013) (ERC Grant 306933 to F.J.R.M.), the Netherlands Organisation for Scientific Research (VICI Grant 016. VICI.170.072 to F.J.R.M.) and Research Foundation Flanders (FWO Aspirant PhD Fellowship to S.V.D.V.). B.J.W.M. is funded by a University of Leeds Academic Fellowship. S.W.P. and T.M.L. acknowledge support from Royal Society Wolfson Research Merit Awards and the NERC Biosphere Evolution, Transitions and Resilience (BETR) programme (NE/P013651).

## Author contributions
S.V.D.V. and F.J.R.M. conceived the hypothesis; S.V.D.V., B.J.W.M. and S.W.P. compiled the data; S.V.D.V. and B.J.W.M. altered the model and performed simulations; T.M.L. provided model code and advice; S.V.D.V., B.J.W.M. and S.W.P. wrote the paper, with significant contributions from all co-authors.

## Additional information

**Competing interests:** The authors declare no competing interests.

