## [Peer Review File · Nature Communications]

Reviewers' comments:

Reviewer #1 (Remarks to the Author):

The authors have revised the COPSE model to include variable bioturbation and investigate the potential impacts that a delayed vs rapid onset of bioturbation might have on a variety of global biogeochemical cycles and the impact on ocean chemistry, redox state, and animal evolution. This represents potentially a quite exciting advance for our understanding of coupled biological/environmental evolution in the Paleozoic. However, the present manuscript had a few key issues that (at least for me) prevent the utility of the particular approach being fully realized.

1) Bioturbation scenarios:

The two scenarios examined however, basically boil down to full bioturbation by the onset of the Cambrian (Scenario 2, Fig 4) or minimal bioturbation during the Cambrian and Ordovician and then rapid increase to full bioturbation in the early Silurian (Scenario 1, Fig 4). These end members are chosen based on different impacts of bioturbation depth on geochemical cycling, which the authors nicely discuss in the text. However, something that doesn't appear to come up is that not only does the intensity of bioturbation increase gradually over the Cambrian-Silurian, but so does its frequency of occurrence/areal extent. It seems like this must matter as well and would serve to make the impact of bioturbation more gradual (something intermediate between scenarios 1-2). I'm surprised it isn't discussed/addressed more clearly in the present submission. It would be nice to have the authors describe exactly what bioturbation data are used to generate the various scenarios used here....

Also, the text refers to modeling anoxia as having no impact on bioturbation vs. resulting in a scaled decrease in bioturbation proportional to the areal extent of anoxia (as discussed in the methods). It would be valuable to see these shown in the figures in the main text.

Also, why does bioturbation in Scenario 2 start 20 Myr before the Cambrian starts, when Scenario 1 has bioturbation begin with the earliest Cambrian? I'm assuming the same set of data are used to determine when bioturbation turns on...

2) Baseline $^{13}\text{C}/^{34}\text{S}$ data:

The curves shown throughout the manuscript don't look like they include the most up to date results and I'm surprised at many of the patterns within them (e.g., my understanding is that sulfate ^{34}S should peak at the Ediacaran-Cambrian boundary with high values through the early Cambrian, before decreasing). Further, these data are interpreted as if the isotopic values can be simply inverted to determine globally averaged organic carbon and/or pyrite burial. While this is a standard simple interpretation, the reality is that these data are much more complex, reflecting multiple drivers (variable microbial fractionations, inputs, etc.) and also impacted in fairly significant ways by local water column conditions, depositional facies, and diagenesis – issues which don't necessarily simply average out.

In sum, while the inclusion of bioturbation to the COPSE model is a valuable addition, the parametrization of the temporal evolution of bioturbation in the present manuscript, the isotopic data against which the model is evaluated, and the remaining discrepancies between them suggest that at present, more work needs to be done to convincingly relate the impact that the expansion of bioturbation has on biogeochemical cycles (and biological evolution) during the first 100+ million years of the Phanerozoic.

If the authors can address these points, then the resulting manuscript could have the potential to provide additional ways to constrain the impact of bioturbation on key biogeochemical cycles and identify interesting potential feedbacks between evolution, ocean chemistry, and extinctions, all of which would be of substantial interest to the community of Earth historians.

Reviewer #3 (Remarks to the Author):

Using a biogeochemical modeling framework (an expanded version of the COPSE model), van de Velde et al. explore the biogeochemical impact of early Paleozoic bioturbation. Specifically, the authors seek to determine, through comparison of model results with contemporaneous proxy records, whether the shallow burrowing and sediment mixing previously documented from lower Paleozoic successions may have had a commensurate (i.e., limited) or disproportionately large impact on sedimentary and global ocean C, P and S cycling (and, by extension, productivity and ocean-atmosphere oxygen levels). With this work, van de Velde et al. shed some much-needed light upon long-standing and critical questions concerning the biogeochemical engineering potential of early bioturbating animals, and the role played by bioturbators in the dynamic environmental conditions characteristic of the early Phanerozoic.

However, with that positive sentiment expressed, I have some concerns with the manuscript. Foremost, the authors' approach—determining, by means of comparison of bioturbation-mediated modeling results with C, S and Ce proxy records, whether early Paleozoic bioturbation had a minor or major impact on contemporaneous biogeochemical cycling—is premised on the assumption that there are no other potential drivers for observed early Paleozoic biogeochemical phenomena (that they are solely the product of bioturbation-mediated perturbation). However, these features and processes of course have a number of potential drivers (for instance, sustained low-oxygen conditions [e.g., in the late Neoproterozoic and early Paleozoic] can be driven by P or N stress), and not solely by bioturbation. The authors' approach certainly has great utility in exploring these issues, but they need to explicitly acknowledge that they are only exploring the role of bioturbation as a driver (which is not the same as testing whether bioturbation or some other factor [or combination of factors] was the principal driver), and that they are not, in this study, examining the role of other and potentially significant factors.

Moreover, there are, in my view, a number of issues with how the authors conceptualize key bioturbation-biogeochemical feedbacks, and with how they employ proxy data to argue for significant feedbacks in the early Paleozoic. For instance, the authors appear to conflate short-term elemental cycling (e.g., S redox oscillations) with longer-term processes (S burial). However, the number of oscillations a S atom undergoes in the sediment pile is insignificant—in terms of geologic time scales and the feedbacks and proxy archives modeled by the authors—relative to the question of how it is ultimately buried (i.e., as pyrite or as sulfate—as tracked by isotope mass balance, e.g. fpyr). Therefore, as changes in the number of redox cycles does not necessarily translate to changes in net burial, it is not clear that the high number of redox oscillations which some of the authors have previously observed to occur at low Db is actually equivalent to the majority of the potential long-term bioturbation-biogeochemical feedback (contra l. 221-232). Moreover, the authors examine the geologic record of S cycling through the lens of $\delta^{34}\text{SSO}_4$ and argue that variations in the Ediacaran–lower Paleozoic $\delta^{34}\text{SSO}_4$ record (l. 111-126) indicate significant shifts in S cycling through this interval. However, in order to make inferences about changes in S burial, these data should be considered in the context of $\delta^{34}\text{Spyr}$ data (e.g., fpyr), and fpyr values actually remain remarkably high and invariant through much of this interval (see Wu et al., 2010, GCA; Leavitt et al., 2013, PNAS).

There are also a number of issues with the authors' treatment of the paleo-O₂ proxy records, in particular their depiction of a linear decrease between the Ediacaran (supposedly higher O₂) and the early Paleozoic (supposedly lower O₂) (l. 102-106, 142-144, 269-278; SI l. 3-9, 18-26). Although it is well-supported (and relatively uncontentious) that oxygen was low, relative to modern concentrations, during the Cambrian and early Ordovician, it is by no means well established (contra SI l. 23-26) that Ediacaran oxygen levels were substantially higher than those

of the early Paleozoic. In fact, numerous studies (including some of those cited by the authors—e.g., Sperling et al., 2015, *Nature*; Wallace et al., 2017, *EPSL*; as well as Sahoo et al., 2012, *Nature*; Sahoo et al., 2016, *Geobiology*; Li et al., 2017, *Geology*), drawing on trace metal, Fe speciation and Ce anomaly data, have in fact stressed the opposite—that Ediacaran oxygen levels were not higher than Cambrian levels, and these studies have indicated significant variability in oxygen levels through this interval. As a side note, the trace metal data presented by Boyle et al. (2014, *Nature Geoscience*) and re-figured by the authors in Fig. S1 include enrichment values far higher than those recorded in modern anoxic basins—which at face value would indicate that early Cambrian oxygen values were higher than today's (and may indicate that these trace metal data are not robust archives of Cambrian anoxia, nor of bioturbation-driven anoxia). Therefore, the authors' preferred Ediacaran-Cambrian decrease is a major, and potentially flawed, oversimplification of the paleo-O₂ records, and conflicts with a number of substantial datasets.

The presented P modeling results (and inferred feedbacks on productivity and ocean-atmosphere oxygen) are intriguing, but as the COPSE model does not include full P speciation, the authors should acknowledge this simplification and the assumptions inherent to their approach. Additionally, given that the proposed bioturbation-CO₂ feedback (e.g., l. 279-289) departs from traditional (e.g., Berner, 2006, *GCA*) views (i.e., that organic carbon burial and remineralization and oxygen are in balance, without impact upon pCO₂), there should be more in-depth discussion of these assumptions and results.

On the whole, I am supportive of the authors' efforts and think that exercises such as theirs provide an important step forward in considering evolutionary-biogeochemical feedbacks through a critical interval of Earth's history. However, I feel strongly that the authors need to reconsider some of the proxy data (i.e., the significant variability in O₂ indicated by Ediacaran and Cambrian proxy data, and joint consideration of $\delta^{34}\text{S}_{\text{SO}_4}$ and $\delta^{34}\text{S}_{\text{pyr}}$ records) and more explicitly discuss the assumptions inherent to their modeling approach.

Minor comments:

l. 16-17: This is rather an odd beginning (particularly as the origin of animals may have preceded the origin of burrowing by 75 million years or more)...consider rephrasing.

l. 26: Here and elsewhere, the "early" of "early Cambrian" should not be capitalized.

l. 38-39: This is certainly not true—many clades of burrowing animals (at the family and even order level) do not appear until the mid-late Paleozoic (or, in some cases, potentially even as late as the Mesozoic).

l. 64-65: Biomixing was also described by the authors as having the opposite trajectory (upward transport) in l. 49-50. Please clarify which direction is considered predominant (and how this is parameterized in the model) and why.

l. 130ff: It should be noted that Tarhan et al. (2015, *Nature Geoscience*) also considered bioturbation to have a non-linear (exponential) relationship with S cycling, on the basis of modern sulfide reoxidation and mixed layer depth data. Therefore the two scenarios should not be depicted as "linear" vs. "non-linear," but rather different exponential relationships. Ultimately, however, it should be explicitly acknowledged that there is currently very little modern data constraining the exact shape of this curve.

l. 175-176: Burial of reduced pyrite will not result in net oxygenation if balanced by pyrite oxidation (delivery via oxidative weathering of continental pyrite). Only if pyrite burial increases and is not matched by increased oxidative input will there be net oxygenation.

l. 197: Please specify to which dotted line this applies (as there are multiple dotted lines).

l. 260-266: As discussed above, it is not the number of oscillations, but the net effect upon burial for which it is important to constrain a relationship with sediment mixing intensity, if the authors' object is to construe processes important on geologic time scales.

l. 274-278: As discussed above, this is premised on the assumption that Cambrian oxygen levels were lower than those of the Ediacaran, which is not robustly supported by the proxy data (as articulated by studies cited by the authors; e.g., Sperling et al., 2015; Wallace et al., 2017). Therefore there are little grounds for considering, under Scenario 1, that rates of organic carbon burial substantially changed as a function of bioturbation between these intervals (i.e., the proxy data indicate that the 'baseline' itself may not have changed [and that the very idea of a stable 'baseline' is questionable] through this interval). Moreover, given that the authors have, as they acknowledge (l. 217-219), chosen to set Silurian $f_{\text{biot}} = 1$ for exploratory purposes alone (as the timing of this increase is speculative), they should not then infer the lack of a match between 'Scenario 1' Silurian model outputs with Silurian $\delta^{13}\text{C}_{\text{carb}}$ to be grounds for rejecting 'Scenario 1'.

l. 290-295: Low $[\text{SO}_4]$, in this light, appears to be a recurring but non-independent solution, as low sulfate has, on the basis of modeling exercises, been suggested to be the product of both a limited bioturbation impact (e.g., Tarhan et al., 2015) and a larger bioturbation impact (this study).

l. 313-320: Here and elsewhere (e.g., Figure S3) please note the meaning of the shading in the panel d, g, h, i and j profiles.

l. 329-331: This variability is also characteristic (although with poorer temporal resolution) of the Ediacaran.

l. 337-339: As discussed above, bioturbation is not the sole potential driver of positive carbon isotope excursions. Productivity collapse (linked to nutrient stress or other environmental perturbations unrelated to bioturbation, for instance) need not (and should not) be attributed to bioturbation alone—this is not a unique solution (as the authors implicitly acknowledge, for S cycling, in l. 546-549).

Response to reviewers.

Van De Velde et al. Nature Communications.

Reviewer's comments are shown in black with our responses in blue. Altered text in the revised manuscript also appears in blue.

Reviewer #1 (Remarks to the Author):

The authors have revised the COPSE model to include variable bioturbation and investigate the potential impacts that a delayed vs rapid onset of bioturbation might have on a variety of global biogeochemical cycles and the impact on ocean chemistry, redox state, and animal evolution. This represents potentially a quite exciting advance for our understanding of coupled biological/environmental evolution in the Paleozoic. However, the present manuscript had a few key issues that (at least for me) prevent the utility of the particular approach being fully realized.

We are grateful to the reviewer for the positive assessment and constructive comments. The revised manuscript is much improved by these comments, and we feel the conclusions are strengthened.

1) Bioturbation scenarios:

The two scenarios examined however, basically boil down to full bioturbation by the onset of the Cambrian (Scenario 2, Fig 4) or minimal bioturbation during the Cambrian and Ordovician and then rapid increase to full bioturbation in the early Silurian (Scenario 1, Fig 4). These end members are chosen based on different impacts of bioturbation depth on geochemical cycling, which the authors nicely discuss in the text. However, something that doesn't appear to come up is that not only does the intensity of bioturbation increase gradually over the Cambrian-Silurian, but so does its frequency of occurrence/areal extent. It seems like this must matter as well and would serve to make the impact of bioturbation more gradual (something intermediate between scenarios 1-2). I'm surprised it isn't discussed/addressed more clearly in the present submission.

This is certainly a valid point, and we have now evaluated an intermediate scenario in the revised paper to more fully represent the range of possibilities (see line 230 and further and revised figures 4 and 5).

Scenario 1 = the biogeochemical impact of bioturbation scales with its intensity and depth; the effect is only ~30% (shallow bioturbation) throughout the Cambrian and Ordovician and rapidly increases to the full impact of bioturbation in the early Silurian

Scenario 2 = shallow bioturbation has a large impact, but that the areal extent of bioturbation (and its effect) gradually increased throughout the early Palaeozoic

Scenario 3 = Shallow bioturbation has a large impact, and was maximal at the start of the Cambrian

The simulation results for scenario 2 plot between those of scenario 1 and 3, as would be expected. When comparing to the available proxy data, we still favour Scenario 3 overall.

It would be nice to have the authors describe exactly what bioturbation data are used to generate the various scenarios used here....

The bioturbation data used are presented in Figure 1d in the current manuscript, and are based on the typical burrow depths described in Herringshaw et al., 2017 and Tarhan et al., 2015, we have now stated this explicitly in the text around line 223.

In addition, we have slightly revised our 'Scenario 1' to more directly link to the observed changes in burrow depth: Scenario 1 now assumes a ~3 fold increase in the burrow depth between Cambrian and Silurian, which is consistent with the data (our previous choice was 10-fold and was likely too extreme). Given the available data, we still find Scenario 3 to generate the most plausible outcome.

Also, the text refers to modeling anoxia as having no impact on bioturbation vs. resulting in a scaled decrease in bioturbation proportional to the areal extent of anoxia (as discussed in the methods). It would be valuable to see these shown in the figures in the main text.

Our apologies, we did use solid and dashed lines in the model outputs to show the effects of including this anoxia feedback, but we accidentally omitted the reference to these lines in the figure captions and text. This has now been amended, and the lines have also been redrawn to be clearer.

Also, why does bioturbation in Scenario 2 start 20 Myr before the Cambrian starts, when Scenario 1 has bioturbation begin with the earliest Cambrian? I'm assuming the same set of data are used to determine when bioturbation turns on...

We have revised these scenarios so that bioturbation now increases between 550 and 540 Ma, such that the maximum value is reached around the Precambrian-Cambrian boundary (see Figure 4).

2) Baseline $^{13}\text{C}/^{34}\text{S}$ data:

The curves shown throughout the manuscript don't look like they include the most up to date results and I'm surprised at many of the patterns within them (e.g., my understanding is that sulfate ^{34}S should peak at the Ediacaran-Cambrian boundary with high values through the early Cambrian, before decreasing).

Our submission used the compilations of Saltzmann & Thomas (2012) and Och and Shields-Zhou (2012). We believe that these datasets are still the most comprehensive and up to date compilations with respect to $d^{13}\text{C}_{\text{carbonate}}$ and $d^{34}\text{S}_{\text{sulphate}}$ values, although of course we welcome any suggested additions.

The LOESS fit we show to the $d^{34}\text{S}$ data is very similar to the fit of Wu et al. (2010), showing a peak around 500 Ma. It is true that more sparse datasets from CAS show a peak in the Early Cambrian, but we do not believe that current CAS datasets (e.g. Algeo et al. 2015) have enough detail for the Palaeozoic to confidently draw this conclusion.

We have changed the y-axis in our plots to show all of the isotope data, so that the pattern is more easily comparable to that shown in other studies. We also note that the plotting of our simulation results versus the CAS dataset would not change our conclusions as none of the model scenarios can generate an Early Cambrian $d^{34}\text{S}$ peak.

Further, these data are interpreted as if the isotopic values can be simply inverted to determine globally averaged organic carbon and/or pyrite burial. While this is a standard simple interpretation, the reality is that these data are much more complex, reflecting multiple drivers (variable microbial fractionations, inputs, etc.) and also impacted in fairly significant ways by local water column conditions, depositional facies, and diagenesis – issues which don't necessarily simply average out.

We agree with this statement, and we have added a paragraph that highlights the limitations, and we also note the other drivers that also can affect the isotope data (see line 334). We also stress that, despite these uncertainties, a ‘forward’ biogeochemical model remains a useful tool for examining possible hypotheses against data compilations, especially over these very long timescales, i.e. it would take a rather specific and unusual set of time-dependent fractionation factors and diagenetic alterations to reverse our conclusions.

In sum, while the inclusion of bioturbation to the COPSE model is a valuable addition, the parametrization of the temporal evolution of bioturbation in the present manuscript, the isotopic data against which the model is evaluated, and the remaining discrepancies between them suggest that at present, more work needs to be done to convincingly relate the impact that the expansion of bioturbation has on biogeochemical cycles (and biological evolution) during the first 100+ million years of the Phanerozoic.

If the authors can address these points, then the resulting manuscript could have the potential to provide additional ways to constrain the impact of bioturbation on key biogeochemical cycles and identify interesting potential feedbacks between evolution, ocean chemistry, and extinctions, all of which would be of substantial interest to the community of Earth historians.

We have addressed all of the above points: we have expanded the estimates for the temporal evolution of bioturbation to cover a broader range of scenarios, we have updated our display of the isotopic data, and we have added a discussion of the remaining discrepancies and their relation to our hypotheses. We believe that the revised paper is much improved by these changes and thank the reviewer for the constructive analysis.

References:

Tarhan et al., 2015, Nature Geoscience 8, 865–869

Herringshaw et al. In: Brasier, A. T., McIlroy, D. & McLoughlin, N. (eds) 2017. Earth System Evolution and Early Life: A Celebration of the Work of Martin Brasier. Geological Society, London, Special

Reviewer #3 (Remarks to the Author):

Using a biogeochemical modeling framework (an expanded version of the COPSE model), van de Velde et al. explore the biogeochemical impact of early Paleozoic bioturbation. Specifically, the authors seek to determine, through comparison of model results with contemporaneous proxy records, whether the shallow burrowing and sediment mixing previously documented from lower Paleozoic successions may have had a commensurate (i.e., limited) or disproportionately large impact on sedimentary and global ocean C, P and S cycling (and, by extension, productivity and ocean-atmosphere oxygen levels). With this work, van de Velde et al. shed some much-needed light upon long-standing and critical questions concerning the biogeochemical engineering potential of early bioturbating animals, and the role played by bioturbators in the dynamic environmental conditions characteristic of the early Phanerozoic.

However, with that positive sentiment expressed, I have some concerns with the manuscript. Foremost, the authors' approach—determining, by means of comparison of bioturbation-mediated modeling results with C, S and Ce proxy records, whether early Paleozoic bioturbation had a minor or major impact on contemporaneous biogeochemical cycling—is premised on the assumption that there are no other potential drivers for observed early Paleozoic biogeochemical phenomena (that they are solely the product of bioturbation-mediated perturbation). However, these features and processes of course have a number of potential drivers (for instance, sustained low-oxygen conditions [e.g., in the late Neoproterozoic and early Paleozoic] can be driven by P or N stress), and not solely by bioturbation. The authors' approach certainly has great utility in exploring these issues, but they need to explicitly acknowledge that they are only exploring the role of bioturbation as a driver (which is not the same as testing whether bioturbation or some other factor [or combination of factors] was the principal driver), and that they are not, in this study, examining the role of other and potentially significant factors.

We agree with this statement and have added text to the revised manuscript (see line 348) to make it clear that there are other factors that can drive Paleozoic environmental change. We also note that the COPSE model does account for some of these drivers – e.g. it has coupled ocean nutrient P and N cycles and it includes changing geological forcing factors over time.

Moreover, there are, in my view, a number of issues with how the authors conceptualize key bioturbation-biogeochemical feedbacks, and with how they employ proxy data to argue for significant feedbacks in the early Paleozoic. For instance, the authors appear to conflate short-term elemental cycling (e.g., S redox oscillations) with longer-term processes (S burial). However, the number of oscillations a S atom undergoes in the sediment pile is insignificant—in terms of geologic time scales and the feedbacks and proxy archives modeled by the authors—relative to the question of how it is ultimately buried (i.e., as pyrite or as sulfate—as tracked by isotope mass balance, e.g. fpyr). Therefore, as **changes in the number of redox cycles does not necessarily translate to changes in net burial**, it is not clear that the high number of redox oscillations which some of the authors have previously observed to occur at low Db is actually equivalent to the majority of the potential long-term bioturbation-biogeochemical feedback (contra l. 221-232).

Our intention here was to use the number of sulfur redox cycles as a rough metric for the ability to preserve reduced species in the sediment. We believe this is reasonable as these redox oscillations are caused by influshing of oxygenated water (bio-irrigation) or upward mixing of reduced sulphur minerals (e.g. iron sulfides) into the oxic zone (bio-mixing). Therefore, the same processes that drive S redox oscillations also ultimately inhibit the rate at which reduced species like pyrite can be buried

over geological timescales. This process is demonstrated in detail by early diagenetic model simulations in van de Velde and Meysman (2017). We acknowledge that this may have been poorly explained, and that this is not a quantitative metric, as the reviewer notes.

We have revised our paper (line 240) to explain this reasoning more carefully, and to make a better case for why we should test the possibility that shallow bioturbation could have had a disproportionate effect on chemical cycling - noting that Nascimento et al. (2012, L&O) have shown that meiofauna (micron-scale animals that burrow ~1cm deep in the sediment) enhance organic carbon breakdown as much as large animals (that burrow > 10 cm deep in the sediment), and that Dale et al. (2015, GCA) use a sediment diagenesis model to show that the levels of bioturbation in the Cambrian could have had a large effect on chemical cycling.

Moreover, the authors examine the geologic record of S cycling through the lens of $\delta^{34}\text{S}_{\text{SO}_4}$ and argue that variations in the Ediacaran–lower Paleozoic $\delta^{34}\text{S}_{\text{SO}_4}$ record (l. 111-126) indicate significant shifts in S cycling through this interval. However, in order to make inferences about changes in S burial, these data should be considered in the context of $\delta^{34}\text{S}_{\text{pyr}}$ data (e.g., fpyr), and fpyr values actually remain remarkably high and invariant through much of this interval (see Wu et al., 2010, GCA; Leavitt et al., 2013, PNAS).

This is a good point, and we agree that the Paleozoic $\delta^{34}\text{S}$ record does not necessarily imply large shifts in the sulphur cycle. Indeed, our model predictions did not produce large shifts in the sulphur cycle either. We have altered the text to be more careful about the interpretation of $\delta^{34}\text{S}$ data, and have added a plot to the SI showing the model f_{pyr} outputs, which are rather invariant and consistent with the papers cited above (see figure SI3).

See line 126 and line 54 in SI.

There are also a number of issues with the authors' treatment of the paleo-O₂ proxy records, in particular their depiction of a linear decrease between the Ediacaran (supposedly higher O₂) and the early Paleozoic (supposedly lower O₂) (l. 102-106, 142-144, 269-278; SI l. 3-9, 18-26). Although it is well-supported (and relatively uncontentious) that oxygen was low, relative to modern concentrations, during the Cambrian and early Ordovician, it is by no means well established (contra SI l. 23-26) that Ediacaran oxygen levels were substantially higher than those of the early Paleozoic. In fact, numerous studies (including some of those cited by the authors—e.g., Sperling et al., 2015, Nature; Wallace et al., 2017, EPSL; as well as Sahoo et al., 2012, Nature; Sahoo et al., 2016, Geobiology; Li et al., 2017, Geology), drawing on trace metal, Fe speciation and Ce anomaly data, have in fact stressed the opposite—that Ediacaran oxygen levels were not higher than Cambrian levels, and these studies have indicated significant variability in oxygen levels through this interval.

This is a useful criticism, and we agree that the deep ocean was not persistently oxygenated for the whole Ediacaran. We have revised our paper to more specifically note that post-Gaskiers (~580 Ma) there was significant oxygenation of parts of the deep ocean, with evidence coming from multiple localities (e.g. Canfield et al. 2007; 2008), but that anoxic conditions had returned by the early Cambrian. We note that this premise is supported by a recent study of selenium isotopes across the Neoproterozoic and into the Cambrian (Pogge von Strandmann et al. , 2015), where progressive oxygenation is apparent across the Ediacaran, with the most expansive and intense oxygenation occurring after the Gaskiers glaciation, precisely when there is evidence for oxygenation of the deep ocean (Canfield et al., 2007; 2008). The selenium isotope study then shows a return to more widespread anoxia in the early Cambrian, which is consistent with independent evidence for a return to anoxic (and often euxinic) conditions at this time (e.g., Kimura and Watanabe, 2001; Canfield et

al., 2008; Goldberg et al., 2005; 2007). This anoxia has then been shown to be pervasive in the Later Cambrian (Gill et al., 2011).

Sperling et al. (2015) find a slight increase in the proportion of anoxic samples between Ediacaran and Cambrian (despite binning into whole-periods), but this study does not really speak to the lateral extent or intensity of deeper ocean oxygenation. Wallace et al. (2017) conclude that there was “a decrease in oxygen levels in the early Cambrian”. Sahoo’s papers concentrate on oxygenation events during the Ediacaran and Cambrian, but do not challenge a general reduction in O₂ between the late Ediacaran and Early Cambrian. Li et al (2017) suggest that the Middle Cambrian witnessed a deepening of oxygenated conditions to mid-depth waters, but this is very different to the deep ocean oxygenation that followed the Gaskiers glaciation. Therefore, while there was clearly spatial heterogeneity in oxygenation across this period, it is deep ocean oxygenation in the Ediacaran and a return to more widespread anoxia in the early Cambrian that is most compellingly demonstrated by the available geochemical data. Thus, the sequence of events we describe is not particularly controversial and is not really challenged by the above cited papers. Nevertheless, we appreciate this comment, and have revised our paper to be more balanced about the nature of Cambrian oxygenation and to note that Cambrian oxygen concentrations may have been quite variable (see line 102).

As a side note, the trace metal data presented by Boyle et al. (2014, Nature Geoscience) and re-figured by the authors in Fig. S1 include enrichment values far higher than those recorded in modern anoxic basins—which at face value would indicate that early Cambrian oxygen values were higher than today’s (and may indicate that these trace metal data are not robust archives of Cambrian anoxia, nor of bioturbation-driven anoxia). Therefore, the authors’ preferred Ediacaran-Cambrian decrease is a major, and potentially flawed, oversimplification of the paleo-O₂ records, and conflicts with a number of substantial datasets.

The most recent trace metal data compilation is from Chen et al. (2015), who argue that the enrichments recorded are in line with modern levels of oxygenation. Furthermore, while there might be some disagreement on the interpretation of the currently available ocean redox proxies, we want to highlight that we employ a multi-proxy forward modelling approach. Our preferred scenario is supported by other proxies ($d^{13}C_{carb}$, $d^{34}S_{SO4}$), and does not only rely on the interpretation of the paleo-O₂ records. We believe therefore that our modelling effort here does help in understanding some of the observed biogeochemical changes in the early Palaeozoic, and is an important step forward.

The presented P modeling results (and inferred feedbacks on productivity and ocean-atmosphere oxygen) are intriguing, but as the COPSE model does not include full P speciation, the authors should acknowledge this simplification and the assumptions inherent to their approach.

We have added text noting that the model P cycle is simplified (as are the other cycles) and stating clearly that the predictions still have a way to go in fully representing the data. See line 352

Additionally, given that the proposed bioturbation-CO₂ feedback (e.g., l. 279-289) departs from traditional (e.g., Berner, 2006, GCA) views (i.e., that organic carbon burial and remineralization and oxygen are in balance, without impact upon pCO₂), there should be more in-depth discussion of these assumptions and results.

We agree that over long timescales the oxygen cycle must be in such a balance, however the changes we see in O₂ levels over the Phanerozoic can only happen through (slight) imbalance in O₂ sources and sinks, and if this is due to changes to organic C burial and weathering, then these changes must also impact CO₂. Both GEOCARBSULF and the COPSE model involve slight imbalances in Organic C burial and weathering, which drive changes in both O₂ and CO₂. See Fig 4 in Shields and Mills (2017, PNAS) for some more information on this.

We have added some text to make this clearer and cited the above paper. (line 315)

On the whole, I am supportive of the authors' efforts and think that exercises such as theirs provide an important step forward in considering evolutionary-biogeochemical feedbacks through a critical interval of Earth's history. However, I feel strongly that the authors need to reconsider some of the proxy data (i.e., the significant variability in O₂ indicated by Ediacaran and Cambrian proxy data, and joint consideration of $\delta^{34}\text{S}_{\text{SO}_4}$ and $\delta^{34}\text{S}_{\text{pyr}}$ records) and more explicitly discuss the assumptions inherent to their modeling approach.

We are happy that the reviewer values our work, and have attempted to fully address their above points. We have added some much-needed discussion of the proxy data and the broad timescale and limitations of our modelling approach. We believe that the revised paper is much improved by these changes and thank the reviewer for their constructive analysis.

References:

- Kimura and Watanba, 2001, *Geology* 29 (11): 995-998.
Goldberg et al., 2005, *Precambrian Research* 137 (3-4): 223-241
Goldberg et al., 2007, *Palaeogeography, Palaeoclimatology, Palaeoecology* 254(1-2):175-193
Gill et al., 2011, *Nature* 469:80-83
Wallace et al., 2017, *Earth and Planetary Science Letters* 466:12-19
Sperling et al., 2015, *Nature* 523:451-454
Canfield et al., 2007, *Science* 315:92-95
Canfield et al., 2008, *Science* 321:949-952
Pogge von Strandmann et al., 2015, *Nature Communications* 6
Sahoo et al., 2012, *Nature* 489:546-549
Sahoo et al., 2016, *Geobiology* 14(5):457-468
Li et al., 2017, *Geology* 45(8):743-746
Dale et al., 2016, *Geochimica et Cosmochimica Acta* 189:251-268
Nascimento et al., 2012, *Limnology and Oceanography* 57(1)338-346.
Boyle et al., 2014, *Nature Geoscience* 7:671-676
Chen et al., 2015, *Nature Communications* 6:7142

Minor comments:

l. 16-17: This is rather an odd beginning (particularly as the origin of animals may have preceded the origin of burrowing by 75 million years or more)...consider rephrasing.

We have rephrased it to *'The evolution of burrowing animals forms a defining moment in the history of the earth'* (L16)

l. 26: Here and elsewhere, the "early" of "early Cambrian" should not be capitalized.

We have changed it in the manuscript.

l. 38-39: This is certainly not true—many clades of burrowing animals (at the family and even order level) do not appear until the mid-late Paleozoic (or, in some cases, potentially even as late as the Mesozoic).

We have rephrased it to *'Burrowing fauna appeared in the Cambrian'* (L37)

l. 64-65: Biomixing was also described by the authors as having the opposite trajectory (upward transport) in l. 49-50. Please clarify which direction is considered predominant (and how this is parameterized in the model) and why.

When considering a 1D sediment column, biomixing has two directions, upwards and downwards. Theoretically, biomixing has a random effect on solid phase particles, and thus neither direction is predominant. However, since we do not consider individual particles, but concentrations of, e.g. reactive phosphorus or iron sulphide, the direction of biomixing, which is considered to be a diffusive process, is towards lower concentrations, which will change depending on the considered element.

We have added “upwards and downwards” to line 35

In the COPSE model, bioturbation is parametrized as having an effect on the burial fluxes, representing the longer term result of mixing and irrigation.

l. 130ff: It should be noted that Tarhan et al. (2015, Nature Geoscience) also considered bioturbation to have a non-linear (exponential) relationship with S cycling, on the basis of modern sulfide reoxidation and mixed layer depth data. Therefore the two scenarios should not be depicted as “linear” vs. “non-linear,” but rather different exponential relationships. Ultimately, however, it should be explicitly acknowledged that there is currently very little modern data constraining the exact shape of this curve.

We have removed the reference to Tarhan et al. (2015) at this instance. However, the conventional view (as described in McIlroy and Logan, 1999, Palaios) is that the effect of bioturbation scales with depth and intensity. As this is also how we have defined our f_{biot} parameter (linearly with the typical burrow depth), we would prefer to keep the linear vs. non-linear terms.

We have added a sentence acknowledging that there is a great lack of data, line 217.

l. 175-176: Burial of reduced pyrite will not result in net oxygenation if balanced by pyrite oxidation (delivery via oxidative weathering of continental pyrite). Only if pyrite burial increases and is not matched by increased oxidative input will there be net oxygenation.

We changed the sentence (L182-183: *'Sulphur cycle. Burial of reduced pyrite results in oxygenation, whereas oxidative weathering of continental pyrite results in de-oxygenation'*)

l. 197: Please specify to which dotted line this applies (as there are multiple dotted lines).

We have specified this.

l. 260-266: As discussed above, it is not the number of oscillations, but the net effect upon burial for which it is important to constrain a relationship with sediment mixing intensity, if the authors' object

is to construe processes important on geologic time scales.

See our reply to the comment above. We believe that the number of oscillations results from the same processes that control the longer term burial rates.

l. 274-278: As discussed above, this is premised on the assumption that Cambrian oxygen levels were lower than those of the Ediacaran, which is not robustly supported by the proxy data (as articulated by studies cited by the authors; e.g., Sperling et al., 2015; Wallace et al., 2017). Therefore there are little grounds for considering, under Scenario 1, that rates of organic carbon burial substantially changed as a function of bioturbation between these intervals (i.e., the proxy data indicate that the 'baseline' itself may not have changed [and that the very idea of a stable 'baseline' is questionable] through this interval).

We argue above that both of these studies, along with many others, support the general trend we advocate.

Moreover, given that the authors have, as they acknowledge (l. 217-219), chosen to set Silurian $f_{\text{biot}} = 1$ for exploratory purposes alone (as the timing of this increase is speculative), they should not then infer the lack of a match between 'Scenario 1' Silurian model outputs with Silurian $\delta^{13}\text{C}_{\text{carb}}$ to be grounds for rejecting 'Scenario 1'.

We emphasise that it is the qualitative change in $\delta^{13}\text{C}$ that does not fit the record here – regardless of the magnitude, an increase in bioturbation effects in the Silurian should cause some reduction in $\delta^{13}\text{C}$, whereas the record shows an increase.

l. 290-295: Low $[\text{SO}_4]$, in this light, appears to be a recurring but non-independent solution, as low sulfate has, on the basis of modeling exercises, been suggested to be the product of both a limited bioturbation impact (e.g., Tarhan et al., 2015) and a larger bioturbation impact (this study).

We have noted this on line 326

l. 313-320: Here and elsewhere (e.g., Figure S3) please note the meaning of the shading in the panel d, g, h, i and j profiles.

We have changed the layout of the figures to remove the need for shading.

l. 329-331: This variability is also characteristic (although with poorer temporal resolution) of the Ediacaran.

We have noted this (line 388)

l. 337-339: As discussed above, bioturbation is not the sole potential driver of positive carbon isotope excursions. Productivity collapse (linked to nutrient stress or other environmental perturbations unrelated to bioturbation, for instance) need not (and should not) be attributed to bioturbation

alone—this is not a unique solution (as the authors implicitly acknowledge, for S cycling, in l. 546-549).

We have added a discussion part here

Reviewers' comments:

Reviewer #1 (Remarks to the Author):

In this revised manuscript, the authors have adapted the COPSE model to include variations in the intensity and onset of bioturbation and investigate the potential impacts that this may have had on a variety of global biogeochemical cycles and on ocean chemistry, redox state, and animal evolution. This represents potentially a quite exciting advance for our understanding of coupled biological/environmental evolution in the Paleozoic. The revisions in this manuscript, particularly the creation of the additional, intermediate bioturbation scenario and more clearly delineated impacts of anoxia-feedbacks represent a substantial improvement.

On a personal note, I fundamentally believe that many of the underlying C and S isotopic data are not representative of the global marine DIC and sulfate reservoirs nor have any bearing of global organic carbon or pyrite burial – and, as such, aren't relevant/useful comparisons for these models. Thus, I appreciate the addition of qualifying text in the revised manuscript highlighting some of the concerns when inverting isotope data to reconstruct global cycling.

In sum, while I have concerns about the fidelity and ultimate meaning of the data being modeled, the approach taken here has value and I believe that the publication of this manuscript will further the conversation in the field about the best ways to reconstruct environmental and ecological change over Earth history.

Reviewer #3 (Remarks to the Author):

Using a biogeochemical modeling framework (an expanded version of the COPSE model), van de Velde et al. explore the biogeochemical impact of early Paleozoic bioturbation. Specifically, the authors seek to determine, through comparison of model results with contemporaneous proxy records, whether the shallow burrowing and sediment mixing previously documented from lower Paleozoic successions may have had a commensurate (i.e., limited) or disproportionately large impact on sedimentary and global ocean C, P and S cycling (and, by extension, productivity and ocean-atmosphere oxygen levels). With this work, van de Velde et al. bring welcome attention to long-standing and critical questions concerning the biogeochemical engineering potential of early bioturbating animals, and the role played by bioturbators in the dynamic environmental conditions characteristic of the early Phanerozoic.

Given that this is a revision, my comments are focused on the authors' responses to previous comments, as well as several lingering issues. The authors' revisions have done much to improve the manuscript. However, I still have some concerns which have not, in my opinion, been adequately addressed by the authors' response letter and revisions. First, there are still a number of issues with the authors' treatment of the paleo-oxygen proxy records—specifically their description of a linear decrease between the Ediacaran (putatively higher O₂) and the early Paleozoic (putatively lower O₂), and their representation of the community as united in this view (e.g., l. 106-109)—a representation which, I would argue, is simply not in line with the current literature and thus the general community view. It has indeed become fairly well accepted that oxygen was low (relative to modern concentrations) during the Cambrian and early Ordovician and that oxygenation of the oceans (particularly the deep oceans) was protracted. However, I maintain that there is not agreement that oxygen unidirectionally or 'broadly' decreased during the late Ediacaran–early Paleozoic. Many recent studies have, in fact, argued the opposite—that oxygenation increased (e.g., in a stepwise manner) across the Neoproterozoic–early Paleozoic transition or during the early–middle Cambrian (e.g., Chen et al., 2014, *Nature Communications*; Pogge von Strandmann et al., 2015, *Nature Communications*; Jin et al., 2016, *EPSL*; Li et al., 2017, *Geology*; Stolper and Keller, 2018, *Nature*). Other studies have, in contrast, indicated no

statistically significant change across the Neoproterozoic–Cambrian transition (e.g., Sperling et al., 2015, *Nature*). Further, other work has stressed extreme (e.g., mixed layer) anoxia in the Ediacaran (e.g., Woods et al., 2015, *Precambrian Research*; Bowyer et al., 2017, *Geobiology*). Many studies (e.g., Canfield et al., 2008, *Science*; Wallace et al., 2017, *EPSL*—and including many of the studies referenced above; e.g., Sperling et al., 2015; Wood et al., 2015; Jin et al., 2016; Bowyer et al. 2017; Li et al., 2017) have emphasized significant redox spatial and temporal heterogeneity. These studies are based on wide range of proxy data and paleoenvironments, and the oxygenation or deoxygenation events they document are commonly not coeval and vary at the basinal scale, as those authors themselves note (e.g., Sperling et al., 2015). Moreover, several of these datasets are based upon proxy data from basinal settings and thus track deep-ocean redox state (e.g., Stolper and Keller, 2018—who argue for progressive oxygenation). The authors should note that the Ce anomaly data for higher Ediacaran oxygen levels (reported in Wallace et al., 2017—the study upon which the authors appear to place the greatest weight) reflect basinal- to regional-scale redox state (due to the short residence time of Ce) in shallow marine settings and that these records are out of sync with the ca. 580 Ma oxygenation event recorded by Fe speciation data in Newfoundland deep-water strata (cf. Canfield et al., 2008) by 30-40 million years. In short, to state that “the available geochemical data” unambiguously indicate a broad pattern of deoxygenation between the Ediacaran and Cambrian is simply a misrepresentation of the literature, which much more resoundingly indicates significant spatial and temporal heterogeneity (at a variety of scales) and perhaps no unambiguous ‘trend’ through this interval. Regardless of the ‘real’ environmental trend in surface oxygen levels, temporal and spatial resolution is likely still too poor, and the data too few (and statistical support lacking) to confidently identify larger-scale patterns with certainty across the Ediacaran-early Paleozoic transition. I hate to seem difficult, but to focus on a subset of data that support a favored hypothesis is not a fair representation of the state of the field. Hopefully future work will continue to improve these records. But, in the meantime, to imbue a particular model for this transition (and arguably the model with, at present, the least support) with false certainty will not help to move forward the geological community’s understanding of this interval. The authors need to significantly adjust their language to explicitly acknowledge the uncertainty associated with their preferred model of progressive deoxygenation (and discuss and cite a broader range of studies—both for and against this view), and this should be done up-front, and not solely toward the end of the paper.

Similarly, although the authors’ expanded discussion of uncertainties (l. 335-358) is a welcome addition to the manuscript, the authors should still, up-front, acknowledge that, in exploring the role of bioturbation, they are explicitly testing only one potential driver for observed early Paleozoic biogeochemical phenomena. Greater frankness at the outset of the paper about the assumptions inherent to this study—both the authors’ inferred pattern of progressive deoxygenation across the Ediacaran–early Paleozoic, and the paper’s focus upon bioturbation alone among potential drivers (explicitly acknowledging that there may be others) would, in my opinion, greatly improve the accessibility and impact of the manuscript.

Additionally, although the authors have usefully expanded their justification for why they consider low intensities of bioturbational mixing to have potentially had a large biogeochemical impact, I am afraid I remain unsatisfied with the authors’ justification for their conflation of short-term elemental cycling (e.g., S redox oscillations) with longer-term processes (S burial). The diagenetic modeling and empirical data presented in van de Velde et al. (2016, *Aquatic Geochemistry*; 2017, *Chemical Geology*) do indeed show that redox oscillations can occur at low Db. However, these data have not been compared (empirically or via diagenetic modeling) to a pyrite/AVS burial flux (further, the sulfide burial flux in the system described by van de Velde et al., 2017 appears to be Fe-limited). Therefore, there is no direct justification for considering the number of redox oscillations a proxy for net reduced S burial (and, by extension, that the geologically relevant biogeochemical ‘maximum response’ may occur at low bioturbation intensities). The authors need to therefore more explicitly acknowledge that this is an assumption which has yet to be verified (e.g., l. 232-234). Additionally, the authors should add their model-output f_{pyr} data (currently in

Fig. S3) to the main text (Fig. 5) and note in the main text that these do not match the empirical fpyr record (e.g., Wu et al., 2010, GCA; Leavitt et al., 2013, PNAS).

The addition of 'scenario 2' (a gradual increase in fbiot) has certainly broadened the scope of this study and the applicability of these results. However, the authors' emphasis upon linearity and inaccurately narrow use of the term "scaling" are likely to cause confusion, particularly as currently used by the authors in reference to the bioturbation literature. "Scaling" can of course take any shape and need not be linear (an exponential or logarithmic scaling is just as much of a 'direct' scaling as a linear scaling). Nor is it justifiable to narrowly attribute interpretation of bioturbation-biogeochemical scaling as linear to the bioturbation literature. McIlroy and Logan (1999, Palaios)—the study cited by the authors as representative of the "conventional view" expressed in their 'Scenario 1' (l. 132-135)—never characterize this 'scaling' as linear. Tarhan et al. (2015, Nature Geoscience) explicitly characterize the relationship between mixed layer depth and sulfide reoxidation as exponential (and increases in bioturbational intensity as gradual, much closer to Scenario 2 than Scenario 1). To set up Scenario 1 as the "conventional view" therefore seems dangerously close to erecting a strawman. I have no objection to the authors' exploration of a linear scaling—I think that this is a very useful modeling exercise—but the authors should revise their discussion of these 'Scenarios' and their model data (l. 219-334) to more accurately reflect that choice of the three scenarios was made simply to explore sensitivity to variation in fbiot, rather than trying to pin these scenarios to views expressed in the early Paleozoic bioturbation literature (this has the effect of trying to force the latter to fit the former). On a related note, the source of the "typical burrow depth" values shown in Fig. 1 (and presumably used to guide choice of fbiot values) remains unclear. Herringshaw et al. (2017, Geol. Soc. Lond. Spec. Publ.) report maximum and 'typical' burrow depths for the Fortunian of Newfoundland. Tarhan et al. (2015) report lower-middle Cambrian burrow depth as commonly mm-scale; Cambro-Ordovician and Ordovician-Silurian: cm-scale, and additionally report maximum burrow depths (which, although within the same order of magnitude, do not fully match the 'typical burrow depth' values chosen by van de Velde et al.). The authors might consider showing mixed layer depths instead of burrow depths (e.g., Tarhan et al., 2015), as the bioturbation processes discussed by the authors and represented by fbiot (e.g., l. 212-217, 231-241) are biodiffusive mixing (whereas the depths of discrete burrows are more important for bioirrigation).

Nonetheless, I remain supportive of the authors' efforts and think that, if they can address these concerns and more openly discuss the assumptions and uncertainties inherent to their approach (including where these agree with or contradict previous studies), this study will be a valuable contribution and help illuminate the interaction between environmental and biological processes over a particularly critical interval in Earth's history.

Minor comments:

l. 16: I appreciate the need for an emphatic beginning, but it is going a little far to characterize complex evolutionary processes as occurring in a "moment".

l. 21: Stratigraphic data, presumably?

l. 83-86: I suggest more clearly parsing out these two separate observations; e.g., "...have shown that not only do biogeochemical processes respond non-linearly to bioturbation, but, moreover, large biogeochemical impacts can occur even at shallow burrowing depths and lower mixing intensities"

l. 117-119: There are of course many perturbations throughout this interval (e.g., the biomeres).

l. 124-126: This sentence is, unfortunately, so vague as to be almost meaningless. Please expand upon this.

l. 131-135, 224-226: The authors' basis for considering burrow depth to have substantially

increased in the Silurian is unclear—this is not shown by Tarhan et al. (2015), and Herringshaw et al.'s (2017) study did not extend beyond lowermost Stage 2 of the Cambrian. However, in carbonate inner shelf settings burrow depth does increase substantially between the Middle and Upper Ordovician (Droser and Bottjer, 1989, *Geology*; Tarhan, 2018, *Earth-Science Reviews*)—though this is not seen in heterolithic siliciclastic successions.

I. 145-147: The authors should take care here to avoid the appearance of circularity. Moreover, this argument appears to rely more on deoxygenation than on low oxygen per se.

I. 181-186: This is somewhat misleading as written (e.g., comparison of processes of delivery to processes of burial) and should be rephrased in terms of burial and redox balance.

I. 223: It might be more accurate to characterize this as “when deeper and more intensively burrowing organisms evolve” (as burrow depth alone is not representative of bioturbation intensity).

I. 227-230: It seems strange to set $f_{biot} = 1$ in the Silurian rather than, for instance, the Devonian, given that Tarhan et al. (2015) and other studies (e.g., Thayer et al., 1974, *Science*) have indicated that Silurian bioturbation remained shallow, whereas major increases in sediment mixing may have begun in the Devonian and Carboniferous. Was the Silurian selected merely to coincide with the interpreted abatement of anoxia and rise in oxygen? Further justification for this choice should be given, given that it conflicts with stratigraphic data.

I. 232: Replace “invoking” with “mediating”. And consider adding “empirical” to this list, given the decades of work that has been done on this topic in places like Long Island Sound (see, for instance, compilation in Fig. S2 and Table S2 of Tarhan et al., 2015).

I. 274-278, 297: As mentioned previously, given that the authors have, as they acknowledge, arbitrarily chosen to set Silurian $f_{biot} = 1$ for exploratory purposes alone, a match or mismatch between Silurian model outputs and Silurian geochemical archives should not be built into evaluation of the feasibility of this (or any other) Scenario.

I. 283: Insert “early Cambrian” before “shallow”.

I. 292-297: Yet this (high organic carbon burial resulting, on geologic time scales, in increased oxygen) is equally true of the Precambrian, which was characterized by both ocean anoxia and a lack of bioturbation.

I. 302-304: As discussed above, it is a misrepresentation of the literature to characterize the Ediacaran-Cambrian transition as experiencing “a strong [and implicitly unidirectional and protracted] increase in ocean anoxia.” And, again, the authors should note that the studies cited here do not suggest a coherent, synchronous pattern. For instance, Canfield et al. (2008) found evidence for transient and regional deep-ocean oxygenation ca. 580 Ma, preceded and followed by ferruginous conditions. Whereas the low Ce/Ce* values reported from the literature by Wallace et al. (2017) are from 550-540 Ma shallow marine successions and Wallace et al. (2017) likewise suggest that this oxygenation was transient and regional. Komura and Watanabe (2001, *Geology*) record redox perturbation at the Precambrian-Cambrian boundary but do not suggest a broader-scale pattern of protracted deoxygenation from an Ediacaran peak to a Cambrian-Ordovician low.

I. 323-325: This is not entirely true; Tarhan et al. (2015), for instance, modified Canfield and Farquhar's (2009, PNAS) model to include gypsum burial at lower sulfate concentrations.

I. 326-327: This should be rephrased to clarify that it is the precise (quantitative) relationship between bioturbation intensity and sulfide reoxidation (in the modern) that remains poorly constrained; the overall positive correlation has, in fact, been well demonstrated.

I. 387-390: This idea that redox-mediated ecosystem restructuring may have mediated the Cambrian Explosion and the GOBE (or even the radiations of the Ediacaran—see Reinhard et al., 2016, PNAS) has been discussed at length by Edwards et al. (2017, Nature Geoscience) and Wood and Erwin (2017, Biological Reviews); these papers should be cited here.

Response to reviewers.

Van De Velde et al. Nature Communications.

We thank the reviewers for their constructive comments, which have allowed us to significantly improve the manuscript. Reviewer's comments are shown in black with our responses in blue. Altered text in the revised manuscript also appears in blue.

Reviewer #1 (Remarks to the Author):

In this revised manuscript, the authors have adapted the COPSE model to include variations in the intensity and onset of bioturbation and investigate the potential impacts that this may have had on a variety of global biogeochemical cycles and on ocean chemistry, redox state, and animal evolution. This represents potentially a quite exciting advance for our understanding of coupled biological/environmental evolution in the Paleozoic. The revisions in this manuscript, particularly the creation of the additional, intermediate bioturbation scenario and more clearly delineated impacts of anoxia-feedbacks represent a substantial improvement.

We are grateful to the reviewer for the positive assessment of our revisions.

On a personal note, I fundamentally believe that many of the underlying C and S isotopic data are not representative of the global marine DIC and sulfate reservoirs nor have any bearing of global organic carbon or pyrite burial – and, as such, aren't relevant/useful comparisons for these models. Thus, I appreciate the addition of qualifying text in the revised manuscript highlighting some of the concerns when inverting isotope data to reconstruct global cycling.

We agree with the reviewer that it is important to keep in mind the limitations of inverting isotopic data. As part of this revision we have extended the discussion related to the uncertainties, and moved it to the model description section (where it fits better). The revised text becomes:

L194-208: *By using COPSE we employ a forward modelling approach, which enables a comparison of model predictions of $\delta^{13}\text{C}_{\text{carb}}$ and $\delta^{34}\text{S}_{\text{SO}_4}$ trends to the independent geological record. This contrasts with inverse modelling, where geological records are used as a model forcing, leaving no potential for quantitative testing of the model results. As with all models, comparison with the geological record requires some assumptions. Foremost, the model predicts $\delta^{13}\text{C}_{\text{carb}}$ and $\delta^{34}\text{S}_{\text{SO}_4}$ trends that are representative of the global marine dissolved inorganic carbon and sulphate reservoirs, reflecting the globally-averaged operation of the long term geochemical cycles (in essence changes in organic carbon and pyrite burial). However, changes in the geological isotope record are not solely dependent on changes in the global biogeochemical cycling of carbon and sulphur, but also incorporate possible effects of diagenesis, or evolutionary changes to the fractionation factors associated with photosynthesis and microbial sulphate reduction. Moreover, it is possible that some data represent regional signals rather than global trends. Nevertheless a quantitative comparison of our model predictions to the geological isotope record provides a useful test of the assumptions underlying our COPSE model.*

In sum, while I have concerns about the fidelity and ultimate meaning of the data being modeled, the approach taken here has value and I believe that the publication of this manuscript will further the conversation in the field about the best ways to reconstruct environmental and ecological change over Earth history.

We thank the reviewer for the overall positive assessment.

Reviewer #3 (Remarks to the Author):

Using a biogeochemical modeling framework (an expanded version of the COPSE model), van de Velde et al. explore the biogeochemical impact of early Paleozoic bioturbation. Specifically, the authors seek to determine, through comparison of model results with contemporaneous proxy records, whether the shallow burrowing and sediment mixing previously documented from lower Paleozoic successions may have had a commensurate (i.e., limited) or disproportionately large impact on sedimentary and global ocean C, P and S cycling (and, by extension, productivity and ocean-atmosphere oxygen levels). With this work, van de Velde et al. bring welcome attention to long-standing and critical questions concerning the biogeochemical engineering potential of early bioturbating animals, and the role played by bioturbators in the dynamic environmental conditions characteristic of the early Phanerozoic.

Given that this is a revision, my comments are focused on the authors' responses to previous comments, as well as several lingering issues. The authors' revisions have done much to improve the manuscript. However, I still have some concerns which have not, in my opinion, been adequately addressed by the authors' response letter and revisions.

We are grateful to the reviewer for the positive assessment of our previous revisions, and have attempted to address all of the remaining concerns in this second revision. We are particularly grateful for the insightful comments on ocean redox chemistry. This has stimulated us to put together a much more rigorous appraisal of the evolution of ocean redox chemistry across the early Palaeozoic (additionally noting that the Ediacaran is not the focus of this study). In this revision, we now also incorporate recently published work which we believe now does show a consistent redox trajectory through the early Cambrian. As such, the manuscript has significantly improved by the expansion on these issues, and we hope that we have satisfied the reviewer with these additions.

First, there are still a number of issues with the authors' treatment of the paleo-oxygen proxy records—specifically their description of a linear decrease between the Ediacaran (putatively higher O₂) and the early Paleozoic (putatively lower O₂), and their representation of the community as united in this view (e.g., l. 106-109)—a representation which, I would argue, is simply not in line with the current literature and thus the general community view. It has indeed become fairly well accepted that oxygen was low (relative to modern concentrations) during the Cambrian and early Ordovician and that oxygenation of the oceans (particularly the deep oceans) was protracted. However, I maintain that there is not agreement that oxygen unidirectionally or 'broadly' decreased during the late Ediacaran–early Paleozoic.

We welcome the opportunity to revisit the reviewers concerns over the proxy record, and indeed, we are grateful for the opportunity to address this issue more fully. In this regard, our evaluation of the proxy record has been significantly aided by two recently published studies (Dahl et al., 2017; Guilbaud et al., 2018). We believe we now have a very solid footing for explaining the trajectory of ocean redox chemistry between the Ediacaran and early Palaeozoic (see below). This has resulted in some important changes to the manuscript, and we believe our model data and interpretations of the proxy record now robustly reflect the most recent developments in this area. The reviewer's comments have made us aware that our description of the oxygenation in the previous version may give rise to misinterpretation. Therefore, we have attempted to describe the oxygen evolution of the

oceans more clearly in the main text. The view that we present is one of a shift to generally more reducing conditions ‘during’ the Cambrian, after the Cambrian explosion (around 520 Ma), rather than a unidirectional decrease in oxygen during the late Ediacaran-early Paleozoic. This was not expressed adequately in our previous version.

We realize that some parts of the previous main text gave the impression of a rapid stepwise O₂ decrease precisely at the Ediacaran-Cambrian boundary. This arises from the formulation of the COPSE model, which is typically used to investigate long timescales and for which the ‘Ediacaran-Cambrian boundary’ is a time period ranging ~600 to ~500 Ma. Such a rapid stepwise drop in O₂ is not consistent with the proxy record. Our revised text and model output is much more careful in the precise timings of these events and we hope that this has resolved the issues.

Many recent studies have, in fact, argued the opposite—that oxygenation increased (e.g., in a stepwise manner) across the Neoproterozoic–early Paleozoic transition or during the early–middle Cambrian (e.g., Chen et al., 2014, *Nature Communications*; Pogge von Strandmann et al., 2015, *Nature Communications*; Jin et al., 2016, *EPSL*; Li et al., 2017, *Geology*; Stolper and Keller, 2018, *Nature*).

We agree entirely with the general thrust here – the Cambrian explosion appears to have coincided with increasing O₂, and this is reflected in our new model output and more nuanced text, which agrees particularly closely with the Chen et al. (2015) study that uses Mo isotopes, and the Dahl et al. (2017) study that uses U isotopes, as global, rather than regional, indicators of the extent of ocean oxygenation. We would, however, dispute the suggestion that the Pogge von Strandmann et al. (2015) paper suggests more oxygenated conditions in the Cambrian. This paper (on which co-author Poulton is an author) presents a very sensitive redox study that clearly highlights progressive oxygenation throughout the Ediacaran, followed by a distinct drop in oxygenation at the Neoproterozoic-Cambrian boundary. There are very little data presented above this boundary to make any inference on subsequent oxygenation through the Cambrian.

We take the opportunity to summarize here our view of the evolution of ocean redox chemistry, as supported by recently published work. There is robust evidence for deep ocean oxygenation in the aftermath of the Gaskiers glaciation (Canfield et al., 2007, 2008). However, this certainly did not occur everywhere. Nevertheless, the Se isotope data of Pogge von Strandmann et al. (2015) convincingly suggest that the Earth system became increasingly oxygenated through the Neoproterozoic, up until the Precambrian-Cambrian boundary. Furthermore, it is absolutely correct that there is wide variability in ocean redox chemistry around the globe during the Ediacaran. The study of Guilbaud et al. (2018), which focusses on the early Cambrian but nevertheless proposes that similar conditions may have existed in the Neoproterozoic, may explain this apparent redox heterogeneity, by the existence of an anoxic oxygen minimum zone above oxygenated deeper waters and below oxygenated shallower waters (as has also been invoked for earlier in the Mesoproterozoic; Zhang et al., 2016). Thus, the precise site of sampling when applying local redox proxies is critical here, which again highlights the importance of more global redox proxies such as Se, Mo and U isotopes. Notwithstanding this, we stress here that modelling the Ediacaran is not the target of this study. We include the Ediacaran in our study merely to highlight the prevailing conditions prior to our more detailed evaluation of the early Cambrian.

There is also abundant evidence for a brief episode of ocean anoxia at the Precambrian-Cambrian boundary (e.g., Canfield et al., 2008; Goldberg et al., 2005; Kimura and Watanabe, 2001; Pogge von

Strandmann et al., 2015). The reason for this is not yet resolved, but again, this short-lived episode is not the focus of the current study, and our modelling techniques are unable to resolve such short-term changes in ocean redox chemistry. More significantly, evidence from Mo isotopes and Mo concentrations suggest that oxygenation of the ocean increased through the early Cambrian up until 520 Ma at the height of the Cambrian explosion (Chen et al., 2015), at which point the global extent of oxygenation reached a maximum before a return to anoxic conditions (as suggested by U isotopes as well as Mo; Dahl et al., 2017), which is also supported by the rare earth element data we reported in the last version of the manuscript (Wallace et al., 2017). Superimposed on this, there is undoubtedly spatial and temporal variability in ocean redox conditions, as would be expected in an inherently 'low oxygen' world. However, Guilbaud et al. (2018) also provide strong evidence for the presence of an anoxic oxygen minimum zone in the earliest Cambrian (as discussed above), with oxygenated deeper waters. Significantly, there is no evidence for this redox structure after 520 Ma in the study of Guilbaud et al. (2018) as there were no deep water samples of this age analysed in this study. Thus, there are now multiple lines of evidence that convincingly argue for progressive oxygenation of the global ocean up until 520 Ma at the height of the Cambrian explosion (but with an anoxic OMZ along productive continental margins), followed by a dramatic decrease in the extent of oxygenation after 520 Ma.

We have attempted to portray this much more clearly in the revised manuscript, and this has also allowed us to more robustly test the response of shallow bioturbation to the compelling history of ocean redox chemistry that now exists.

Other studies have, in contrast, indicated no statistically significant change across the Neoproterozoic–Cambrian transition (e.g., Sperling et al., 2015, *Nature*).

Like many in the community, we have problems with the suggestion by Sperling et al (2015) that Fe speciation can be used to identify no statistical change in oxygenation across the Neoproterozoic–Cambrian transition. Fe speciation reflects an on-off switch, whereby Fe enrichments cease as soon as there is a small amount of oxygen in the water column (see the comparison of dysoxic and oxic marine sediments in Raiswell and Canfield (1998)). Thus, the Fe speciation proxy says nothing about rising or falling oxygen above a very low oxygen threshold. Indeed, the data of Sperling et al. (2015), which are inherently regional in context and focussed on shelf settings that might be expected to be biased towards anoxia, directly contradict the more sensitive redox proxy data of various studies, including the Chen et al. (2015) study which is far less regional in context. In addition, the study of Sperling et al. (2015) is undoubtedly compromised by (unintentional) biased sampling of shelf environments (c.f. Guilbaud et al., 2018).

Further, other work has stressed extreme (e.g., mixed layer) anoxia in the Ediacaran (e.g., Woods et al., 2015, *Precambrian Research*; Bowyer et al., 2017, *Geobiology*).

See our response above. We would agree entirely that mixed layer anoxia was prevalent throughout this time period, but again stress that our focus is on the early Cambrian.

Many studies (e.g., Canfield et al., 2008, *Science*; Wallace et al., 2017, *EPSL*—and including many of the studies referenced above; e.g., Sperling et al., 2015; Wood et al., 2015; Jin et al., 2016; Bowyer et al. 2017; Li et al., 2017) have emphasized significant redox spatial and temporal heterogeneity.

We agree entirely – see response above.

These studies are based on wide range of proxy data and paleoenvironments, and the oxygenation or deoxygenation events they document are commonly not coeval and vary at the basinal scale, as those authors themselves note (e.g., Sperling et al., 2015). Moreover, several of these datasets are based upon proxy data from basinal settings and thus track deep-ocean redox state (e.g., Stolper and Keller, 2018—who argue for progressive oxygenation).

The dataset of Stolper and Keller (2018) (published after our submission) shows a decreasing trend in deep water O₂ with time for the Early Palaeozoic. The ‘progressive oxygenation’ statement applies over longer timescales e.g. whole Paleozoic. Due to the sparse nature of the data, it is a stretch to say it unambiguously supports our view, but it certainly does not contradict it.

The authors should note that the Ce anomaly data for higher Ediacaran oxygen levels (reported in Wallace et al., 2017—the study upon which the authors appear to place the greatest weight) reflect basinal- to regional-scale redox state (due to the short residence time of Ce) in shallow marine settings and that these records are out of sync with the ca. 580 Ma oxygenation event recorded by Fe speciation data in Newfoundland deep-water strata (cf. Canfield et al., 2008) by 30-40 million years.

We hope we have now cleared up this issue by carefully stating that we expect high O₂ around 540 – 521 Ma and a drop afterwards, exactly as shown in the Wallace et al. (2017) paper. We have also reduced our reliance on this dataset as described below. Indeed, this is now one of multiple independent lines of evidence in support of the redox evolution that we describe.

In short, to state that “the available geochemical data” unambiguously indicate a broad pattern of deoxygenation between the Ediacaran and Cambrian is simply a misrepresentation of the literature, which much more resoundingly indicates significant spatial and temporal heterogeneity (at a variety of scales) and perhaps no unambiguous ‘trend’ through this interval.

We agree with the above statement and are grateful for the insight that our description of the oxygenation in the previous version could lead to misinterpretation . We have made careful adjustments to our paper to prevent it being interpreted in this way. We have made it clear that our argument is that a number of proxies support ocean deoxygenation following the Cambrian explosion (L107-140).

Regardless of the 'real' environmental trend in surface oxygen levels, temporal and spatial resolution is likely still too poor, and the data too few (and statistical support lacking) to confidently identify larger-scale patterns with certainty across the Ediacaran-early Paleozoic transition. I hate to seem difficult, but to focus on a subset of data that support a favored hypothesis is not a fair representation of the state of the field. Hopefully future work will continue to improve these records. But, in the meantime, to imbue a particular model for this transition (and arguably the model with, at present, the least support) with false certainty will not help to move forward the geological community’s understanding of this interval. The authors need to significantly adjust their language to explicitly acknowledge the uncertainty associated with their preferred model of progressive deoxygenation (and discuss and cite a broader range of studies—both for and against this view), and this should be done up-front, and not solely toward the end of the paper.

Overall we agree with the reviewer here and hope that our revised manuscript conveys this effectively. We have made our position clearer and noted explicitly that deoxygenation followed the Cambrian explosion. We have included additional discussion about the heterogeneity of redox proxies, and have pictured sedimentary Mo data in the revised manuscript instead of the Ce anomaly, which better represents the global oxygenation state of the ocean, but we also note that the deoxygenation evident in the Ce record after the Cambrian explosion is also entirely consistent with our arguments, as discussed in detail above.

Similarly, although the authors' expanded discussion of uncertainties (l. 335-358) is a welcome addition to the manuscript, the authors should still, up-front, acknowledge that, in exploring the role of bioturbation, they are explicitly testing only one potential driver for observed early Paleozoic biogeochemical phenomena. Greater frankness at the outset of the paper about the assumptions inherent to this study—both the authors' inferred pattern of progressive deoxygenation across the Ediacaran–early Paleozoic, and the paper's focus upon bioturbation alone among potential drivers (explicitly acknowledging that there may be others) would, in my opinion, greatly improve the accessibility and impact of the manuscript.

We have modified the introduction text to note that bioturbation is “a significant potential driver of environmental change”, and have added text directing the reader to the overview of the COPSE model for further information on the processes and drivers already included (L188-189).

Additionally, although the authors have usefully expanded their justification for why they consider low intensities of bioturbational mixing to have potentially had a large biogeochemical impact, I am afraid I remain unsatisfied with the authors' justification for their conflation of short-term elemental cycling (e.g., S redox oscillations) with longer-term processes (S burial). The diagenetic modeling and empirical data presented in van de Velde et al. (2016, Aquatic Geochemistry; 2017, Chemical Geology) do indeed show that redox oscillations can occur at low Db. However, these data have not been compared (empirically or via diagenetic modeling) to a pyrite/AVS burial flux (further, the sulfide burial flux in the system described by van de Velde et al., 2017 appears to be Fe-limited). Therefore, there is no direct justification for considering the number of redox oscillations a proxy for net reduced S burial (and, by extension, that the geologically relevant biogeochemical 'maximum response' may occur at low bioturbation intensities). The authors need to therefore more explicitly acknowledge that this is an assumption which has yet to be verified (e.g., l. 232-234).

We have now made clear that this is an assumption, which has roots in theory and modelling. (e.g. “we suggest”, “proposed” and “potentially” have been added). Indeed, the hypothesis that low levels of bioturbation cause large changes in geochemistry is something we put forward in this paper, and not an idea that has been around in the literature, although we note that Berner and Westrich (1985) and Canfield and Farquhar (2009) argued that the evolution of bioturbation possibly caused a decrease in pyrite burial by increasing the re-oxidation (e.g. redox cycling) of sulphur (L266-269).

Additionally, the authors should add their model-output f_{pyr} data (currently in Fig. S3) to the main text (Fig. 5) and note in the main text that these do not match the empirical f_{pyr} record (e.g., Wu et al., 2010, GCA; Leavitt et al., 2013, PNAS).

As requested we have included the model f_{pyr} output in the manuscript in all figures (Fig.1, Fig. 3 and Fig. 5), and compared it to the range denoted by Wu et al. (2010), in their figure 8i. The f_{pyr} error

range is extremely large (one of the discussion points of their paper), and our results plot within this. If the reviewer can clarify how our results disagree with the empirical record then we are happy to incorporate this in a further revision.

The addition of 'scenario 2' (a gradual increase in f_{biot}) has certainly broadened the scope of this study and the applicability of these results. However, the authors' emphasis upon linearity and inaccurately narrow use of the term "scaling" are likely to cause confusion, particularly as currently used by the authors in reference to the bioturbation literature. "Scaling" can of course take any shape and need not be linear (an exponential or logarithmic scaling is just as much of a 'direct' scaling as a linear scaling).

We have changed "direct scaling" to "direct linear scaling".

Nor is it justifiable to narrowly attribute interpretation of bioturbation-biogeochemical scaling as linear to the bioturbation literature. McIlroy and Logan (1999, Palaios)—the study cited by the authors as representative of the "conventional view" expressed in their 'Scenario 1' (l. 132-135)—never characterize this 'scaling' as linear. Tarhan et al. (2015, Nature Geoscience) explicitly characterize the relationship between mixed layer depth and sulfide reoxidation as exponential (and increases in bioturbational intensity as gradual, much closer to Scenario 2 than Scenario 1). To set up Scenario 1 as the "conventional view" therefore seems dangerously close to erecting a strawman. I have no objection to the authors' exploration of a linear scaling—I think that this is a very useful modeling exercise—but the authors should revise their discussion of these 'Scenarios' and their model data (l. 219-334) to more accurately reflect that choice of the three scenarios was made simply to explore sensitivity to variation in f_{biot}, rather than trying to pin these scenarios to views expressed in the early Paleozoic bioturbation literature (this has the effect of trying to force the latter to fit the former).

We have been more careful to present our scenarios as an envelope of possibilities rather than reflecting exactly previous interpretations. We have still made the point at the beginning of the results section that different interpretations of the biogeochemical effects of bioturbation exist, and are part of the premise for this study, but have revised it significantly:

(L154-160) "Various scalings of the biogeochemical response to bioturbation have been shown to be consistent with subsets of available geochemical proxies: a rapid and non-linear response at low bioturbation intensities may have increased phosphate retention in the sediment when the first shallow-burrowing animals appeared in the Cambrian (Boyle et al., 2014), leading to a decrease in the oxygen production source from organic carbon burial, and driving a return to anoxic ocean conditions after 520 Ma (Fig. 1a). In contrast, a more protracted response has been shown to be consistent with low sulphate concentrations throughout the Palaeozoic (Tarhan et al., 2015)"

On a related note, the source of the "typical burrow depth" values shown in Fig. 1 (and presumably used to guide choice of f_{biot} values) remains unclear. Herringshaw et al. (2017, Geol. Soc. Lond. Spec. Publ.) report maximum and 'typical' burrow depths for the Fortunian of Newfoundland. Tarhan et al. (2015) report lower–middle Cambrian burrow depth as commonly mm-scale; Cambro-Ordovician and Ordovician–Silurian: cm-scale, and additionally report maximum burrow depths (which, although within the same order of magnitude, do not fully match the 'typical burrow depth' values chosen by van de Velde et al.). The authors might consider showing mixed layer depths instead of burrow

depths (e.g., Tarhan et al., 2015), as the bioturbation processes discussed by the authors and represented by fbiot (e.g., l. 212-217, 231-241) are biodiffusive mixing (whereas the depths of discrete burrows are more important for bioirrigation).

We have now changed Fig. 1, and now show the mixed later depth, as presented in Tarhan et al., (2015).

Nonetheless, I remain supportive of the authors' efforts and think that, if they can address these concerns and more openly discuss the assumptions and uncertainties inherent to their approach (including where these agree with or contradict previous studies), this study will be a valuable contribution and help illuminate the interaction between environmental and biological processes over a particularly critical interval in Earth's history.

We thank the reviewer for the detailed review, and hope that the additional revisions have addressed the concerns. We certainly believe that we now have a much stronger manuscript that better matches the complex spatial and temporal variability that is current inherent in the available data.

Minor comments:

l. 16: I appreciate the need for an emphatic beginning, but it is going a little far to characterize complex evolutionary processes as occurring in a "moment".

We changed this to 'event'

l. 21: Stratigraphic data, presumably?

'Stratigraphic data' might be considered to represent data used to define stratigraphy. Thus we have altered the text to 'contemporaneous geochemical data'.

l. 83-86: I suggest more clearly parsing out these two separate observations; e.g., "...have shown that not only do biogeochemical processes respond non-linearly to bioturbation, but, moreover, large biogeochemical impacts can occur even at shallow burrowing depths and lower mixing intensities"

We have altered the text as suggested.

l. 117-119: There are of course many perturbations throughout this interval (e.g., the biomes).

We have amended to "long term perturbation", the subsequent text noting shorter timescale fluctuations also remains.

l. 124-126: This sentence is, unfortunately, so vague as to be almost meaningless. Please expand upon this.

We have rephrased this sentence.

l. 131-135, 224-226: The authors' basis for considering burrow depth to have substantially increased in the Silurian is unclear—this is not shown by Tarhan et al. (2015), and Herringshaw et al.'s (2017) study did not extend beyond lowermost Stage 2 of the Cambrian. However, in carbonate inner shelf

settings burrow depth does increase substantially between the Middle and Upper Ordovician (Droser and Bottjer, 1989, *Geology*; Tarhan, 2018, *Earth-Science Reviews*)—though this is not seen in heterolithic siliciclastic successions.

We have replaced the burrow depths with mixed layer depths following a previous comment, these follow exactly Tarhan et al. (2015).

I. 145-147: The authors should take care here to avoid the appearance of circularity. Moreover, this argument appears to rely more on deoxygenation than on low oxygen per se.

We have simplified this sentence to make a single clear point that pyrite burial rates, as calculated from $d^{34}\text{S}$, are very uncertain.

I. 181-186: This is somewhat misleading as written (e.g., comparison of processes of delivery to processes of burial) and should be rephrased in terms of burial and redox balance.

We have added text on the redox balance to this caption. We are unsure what is misleading about describing the phosphorus inputs and outputs: Riverine P delivery and sedimentary P burial define the marine phosphorus budget and are commonly compared over long timescales e.g. (Compton et al., 2000; Paytan and McLaughlin, 2007).

I. 223: It might be more accurate to characterize this as “when deeper and more intensively burrowing organisms evolve” (as burrow depth alone is not representative of bioturbation intensity).

We have rephrased accordingly.

I. 227-230: It seems strange to set $f_{\text{biot}} = 1$ in the Silurian rather than, for instance, the Devonian, given that Tarhan et al. (2015) and other studies (e.g., Thayer et al., 1974, *Science*) have indicated that Silurian bioturbation remained shallow, whereas major increases in sediment mixing may have begun in the Devonian and Carboniferous.

We have reconsidered the trajectory of f_{biot} to reach a maximum in the Devonian as suggested. Our conclusions are unaffected.

Was the Silurian selected merely to coincide with the interpreted abatement of anoxia and rise in oxygen? Further justification for this choice should be given, given that it conflicts with stratigraphic data.

It was not. The Silurian was selected initially to conform to the phrase “the mixing of sediments on marine shelves remained limited until at least the late Silurian” in the abstract of Tarhan et al. (2015) We have revised this to Devonian in line with the reviewer’s comments, with no change in conclusions.

I. 232: Replace “invoking” with “mediating”. And consider adding “empirical” to this list, given the decades of work that has been done on this topic in places like Long Island Sound (see, for instance, compilation in Fig. S2 and Table S2 of Tarhan et al., 2015).

We have changed invoking to mediating. We do believe that experimental work implies empirical work.

I. 274-278, 297: As mentioned previously, given that the authors have, as they acknowledge, arbitrarily chosen to set Silurian $f_{\text{biot}} = 1$ for exploratory purposes alone, a match or mismatch between Silurian model outputs and Silurian geochemical archives should not be built into evaluation of the feasibility of this (or any other) Scenario.

We have now set $f_{\text{biot}}=1$ in the Devonian in line with these comments. We have also revised the model results text in line with the slightly revised model. A key part of the work is comparing the model results to geochemical archives, and this is altered but must remain. For scenario 1:

(L331-334) “Both of these predictions are to some extent at odds with the geochemical record, which shows evidence for significant ocean anoxia just after the Cambrian Explosion (520 Ma) (Dahl et al., 2017) until the middle Ordovician (470-460 Ma) (Thompson and Kah, 2012), and does not support high $\delta^{13}\text{C}_{\text{carb}}$ ($\sim 2\text{‰}$) throughout this interval (Fig. 1)”

I. 283: Insert “early Cambrian” before “shallow”.

Altered as suggested.

I. 292-297: Yet this (high organic carbon burial resulting, on geologic time scales, in increased oxygen) is equally true of the Precambrian, which was characterized by both ocean anoxia and a lack of bioturbation.

In this manuscript, we try to focus on the impact of bioturbation on the biogeochemical cycles of the early Palaeozoic. The controls on Precambrian oxygen and ocean anoxia are a separate subject, and are still a matter of much debate (see e.g. Lenton and Daines, 2017). Most likely, these mechanisms involve earlier changes in tectonic drivers and biological drivers, such as the onset of a eukaryotic biological pump. So, to explain the high rates of organic carbon burial (which may not be correct) and low levels of oxygen in the Precambrian, one has to invoke different mechanisms. Therefore, we see this as a separate issue than what is discussed in this manuscript.

I. 302-304: As discussed above, it is a misrepresentation of the literature to characterize the Ediacaran-Cambrian transition as experiencing “a strong [and implicitly unidirectional and protracted] increase in ocean anoxia.” And, again, the authors should note that the studies cited here do not suggest a coherent, synchronous pattern. For instance, Canfield et al. (2008) found evidence for transient and regional deep-ocean oxygenation ca. 580 Ma, preceded and followed by ferruginous conditions. Whereas the low Ce/Ce* values reported from the literature by Wallace et al. (2017) are from 550-540 Ma shallow marine successions and Wallace et al. (2017) likewise suggest that this oxygenation was transient and regional. Komura and Watanabe (2001, Geology) record redox perturbation at the Precambrian-Cambrian boundary but do not suggest a broader-scale pattern of protracted deoxygenation from an Ediacaran peak to a Cambrian–Ordovician low.

Following our discussion previously we have changed this passage to ‘an increase in ocean anoxia following the Cambrian explosion’

l. 323-325: This is not entirely true; Tarhan et al. (2015), for instance, modified Canfield and Farquhar's (2009, PNAS) model to include gypsum burial at lower sulfate concentrations.

We have rephrased this to make clear we are only talking about the paper by Canfield and Farquhar (2009).

l. 326-327: This should be rephrased to clarify that it is the precise (quantitative) relationship between bioturbation intensity and sulfide reoxidation (in the modern) that remains poorly constrained; the overall positive correlation has, in fact, been well demonstrated.

We have rephrased it so it is clear that we are talking about the relation between the ocean sulphate inventory and the evolution of bioturbation that is uncertain, since we find no clear effect, in contrast to Canfield and Farquhar who find a clear increase in ocean sulphate concentrations

l. 387-390: This idea that redox-mediated ecosystem restructuring may have mediated the Cambrian Explosion and the GOBE (or even the radiations of the Ediacaran—see Reinhard et al., 2016, PNAS) has been discussed at length by Edwards et al. (2017, Nature Geoscience) and Wood and Erwin (2017, Biological Reviews); these papers should be cited here.

We have included the proposed references

References:

- Berner, R. A., and Westrich, J. T. (1985). Bioturbation and the early diagenesis of carbon and sulfur. *Am. J. Sci.* 285, 193–206.
- Boyle, R. A., Dahl, T. W., Dale, A. W., Zhu, M., Brasier, M. D., Canfield, D. E., et al. (2014). Stabilization of the coupled oxygen and phosphorus cycles by the evolution of bioturbation. *Nat. Geosci.* 7, 671–676. doi:10.1038/NGEO2213.
- Canfield, D. E., and Farquhar, J. (2009). Animal evolution, bioturbation, and the sulfate concentration of the oceans. *Proc. Natl. Acad. Sci.* 106, 8123–8127.
- Canfield, D. E., Poulton, S. W., Knoll, A. H., Narbonne, G. M., Ross, G., Goldberg, T., et al. (2008). Ferruginous conditions dominated Neoproterozoic Deep-Water Chemistry. *Science* (80-.). 321, 949–952.
- Canfield, D. E., Poulton, S. W., and Narbonne, G. M. (2007). Late-Neoproterozoic deep-ocean oxygenation and the rise of animal life. *Science* 315, 92–95. doi:10.1126/science.1135013.
- Chen, X., Ling, H.-F., Vance, D., Shields-Zhou, G. a., Zhu, M., Poulton, S. W., et al. (2015). Rise to modern levels of ocean oxygenation coincided with the Cambrian radiation of animals. *Nat. Commun.* 6, 7142. doi:10.1038/ncomms8142.
- Compton, J., Mallinson, D., Glenn, C. R., Filippelli, G., Föllmi, K., Shields, G., et al. (2000). Variations in the Global Phosphorus Cycle. Available at: http://archives.datapages.com/data/sepm_sp/SP66/Variations_in_the_Global_Phosphorus_Cycle.htm [Accessed April 23, 2018].
- Dahl, T. W., Connelly, J. N., Kouchinsky, A., Gill, B. C., Månsson, S. F., and Bizzarro, M. (2017). Reorganisation of Earth's biogeochemical cycles briefly oxygenated the oceans 520 Myr ago. *Geochemical Perspect. Lett.*, 210–220. doi:10.7185/geochemlet.1724.
- Goldberg, T., Poulton, S. W., and Strauss, H. (2005). Sulphur and oxygen isotope signatures of late Neoproterozoic to early Cambrian sulphate, Yangtze Platform, China: Diagenetic constraints and seawater evolution. *Precambrian Res.* 137, 223–241. doi:10.1016/J.PRECAMRES.2005.03.003.
- Guilbaud, R., Slater, B. J., Poulton, S. W., Harvey, T. H. P., Brocks, J. J., Nettersheim, B. J., et al. (2018).

- Oxygen minimum zones in the early Cambrian ocean. *Geochemical Perspect. Lett.* 6, 33–38. doi:10.7185/geochemlet.1806.
- Kimura, H., and Watanabe, Y. (2001). Oceanic anoxia at the Precambrian-Cambrian boundary. *Geology* 29, 995. doi:10.1130/0091-7613(2001)029<0995:OAATPC>2.0.CO;2.
- Lenton, T. M., and Daines, S. J. (2017). Biogeochemical Transformations in the History of the Ocean. *Ann. Rev. Mar. Sci.* 9, 31–58. doi:10.1146/annurev-marine-010816-060521.
- Paytan, A., and McLaughlin, K. (2007). The Oceanic Phosphorus Cycle. *Chem. Rev.* 107, 563–576. doi:10.1021/CR0503613.
- Pogge von Strandmann, P. A. E., Stüeken, E. E., Elliott, T., Poulton, S. W., Dehler, C. M., Canfield, D. E., et al. (2015). Selenium isotope evidence for progressive oxidation of the Neoproterozoic biosphere. *Nat. Commun.* 6, 10157. doi:10.1038/ncomms10157.
- Raiswell, R., and Canfield, D. E. (1998). Sources of iron for pyrite formation in marine sediments. *Am. J. Sci.* 298, 219–245. doi:10.2475/ajs.298.3.219.
- Sperling, E. a., Wolock, C. J., Morgan, A. S., Gill, B. C., Kunzmann, M., Halverson, G. P., et al. (2015). Statistical analysis of iron geochemical data suggests limited late Proterozoic oxygenation. *Nature* 523, 451–454. doi:10.1038/nature14589.
- Stolper, D. A., and Keller, C. B. (2018). A record of deep-ocean dissolved O₂ from the oxidation state of iron in submarine basalts. *Nature* 3. doi:10.1038/nature25009.
- Tarhan, L. G., Droser, M. L., Planavsky, N. J., and Johnston, D. T. (2015). Protracted development of bioturbation through the early Palaeozoic Era. *Nat. Geosci.* 8, 865–869. doi:10.1038/ngeo2537.
- Thompson, C. K., and Kah, L. C. (2012). Sulfur isotope evidence for widespread euxinia and a fluctuating oxycline in Early to Middle Ordovician greenhouse oceans. *Palaeogeogr. Palaeoclimatol. Palaeoecol.* 313–314, 189–214. doi:10.1016/j.palaeo.2011.10.020.
- Wallace, M. W., Hood, A. vS., Shuster, A., Greig, A., Planavsky, N. J., and Reed, C. P. (2017). Oxygenation history of the Neoproterozoic to early Phanerozoic and the rise of land plants. *Earth Planet. Sci. Lett.* 466, 12–19. doi:10.1016/j.epsl.2017.02.046.
- Wu, N., Farquhar, J., Strauss, H., Kim, S. T., and Canfield, D. E. (2010). Evaluating the S-isotope fractionation associated with Phanerozoic pyrite burial. *Geochim. Cosmochim. Acta* 74, 2053–2071. doi:10.1016/j.gca.2009.12.012.
- Zhang, S., Wang, X., Wang, H., Bjerrum, C. J., Hammarlund, E. U., Costa, M. M., et al. (2016). Sufficient oxygen for animal respiration 1,400 million years ago. *Proc. Natl. Acad. Sci. U. S. A.* 113, 1731–6. doi:10.1073/pnas.1523449113.

REVIEWERS' COMMENTS:

Reviewer #3 (Remarks to the Author):

Van de Velde et al. use COPSE to assess the biogeochemical impact of early Paleozoic bioturbation. In this framework, the authors explore whether the shallow burrowing and sediment mixing previously documented from lower Paleozoic successions were likely to have had a commensurate or large impact on sedimentary and global ocean C, P and S cycling (and, by extension, productivity and ocean-atmosphere oxygen levels). With this work, van de Velde et al. bring welcome attention to long-standing and critical questions concerning the biogeochemical engineering potential of early bioturbating animals, and the role played by bioturbators in the dynamic environmental conditions characteristic of the early Phanerozoic.

This revised version of the manuscript, and the authors' responses to my previous comments have now addressed nearly all of my concerns, and I am fully supportive of publication. My remaining concerns are relatively minor, and I am confident that these can be easily addressed without additional review.

Thank you for adding the model-output fpyr data (and record-based fpyr data) to the main text figures. To clarify my previous comment, I would suggest that the authors note which fpyr estimates and/or methods they used, as there is much larger range of estimates in Wu et al. (2010, GCA, fig. 8) than is displayed here.

I. 81: A little more moderation would be advisable here—it is something of an overstatement to say that bioturbation has previously been considered to impact biogeochemistry ONLY under modern intensities of the former. I would replace “only” with “more” or “especially.”

I. 84: Replace “modern day levels” with “major increases in” (as the full rise to modern-style mixing intensities may have postdated the Devonian).

I. 88: As above, more circumspection would be appropriate here, given the limited number of systems where this has been explicitly explored, and lingering uncertainties for even those systems which have been explored; replace “the most drastic changes” with “some of the most drastic changes.”

I. 133: Hyphenate “near-modern” (compound adjective preceding a noun).

I. 333, 340-341: I suggest changing “after the Cambrian Explosion” to “in the early Cambrian” as, by the standards of some archives (e.g., the non-SSF body fossil record) the Cambrian Explosion does not even start until ca. 520 Ma.

Fig. 1, 3, 5: Check that only the euxinic subset of the Boyle et al. (2014) compilation of Mo concentration data are used here, as euxinic sediments should not be compared to non-euxinic sediments (given different potentials for trace metal enrichment). Also, there have been a considerable number of Mo studies of this time interval since publication of the compilation of Boyle et al. (2014) (e.g., Jin et al., 2016, EPSL, among many others). It would be good to add these recent data.

Response to reviewers.

Van De Velde et al. Nature Communications.

We thank the reviewers for their constructive comments, which have allowed us to significantly improve the manuscript. Reviewer's comments are shown in black with our responses in blue. Altered text in the revised manuscript also appears in blue.

Reviewer #3 (Remarks to the Author):

Van de Velde et al. use COPSE to assess the biogeochemical impact of early Paleozoic bioturbation. In this framework, the authors explore whether the shallow burrowing and sediment mixing previously documented from lower Paleozoic successions were likely to have had a commensurate or large impact on sedimentary and global ocean C, P and S cycling (and, by extension, productivity and ocean-atmosphere oxygen levels). With this work, van de Velde et al. bring welcome attention to long-standing and critical questions concerning the biogeochemical engineering potential of early bioturbating animals, and the role played by bioturbators in the dynamic environmental conditions characteristic of the early Phanerozoic.

This revised version of the manuscript, and the authors' responses to my previous comments have now addressed nearly all of my concerns, and I am fully supportive of publication. My remaining concerns are relatively minor, and I am confident that these can be easily addressed without additional review.

We are grateful to the reviewer for the positive assessment of our previous revisions, and are happy that (s)he agrees with our efforts. We have addressed the last final concerns below.

Thank you for adding the model-output fpyr data (and record-based fpyr data) to the main text figures. To clarify my previous comment, I would suggest that the authors note which fpyr estimates and/or methods they used, as there is much larger range of estimates in Wu et al. (2010, GCA, fig. 8) than is displayed here.

The range we displayed is the full range displayed in Wu et al. (Fig. 8I). We have added a line in the Figure caption which says we are showing the full range.

I. 81: A little more moderation would be advisable here—it is something of an overstatement to say that bioturbation has previously been considered to impact biogeochemistry ONLY under modern intensities of the former. I would replace “only” with “more” or “especially.”

We have replaced only with more.

I. 84: Replace “modern day levels” with “major increases in” (as the full rise to modern-style mixing intensities may have postdated the Devonian).

We have updated the sentence to ‘major increases in burrowing occurred’

I. 88: As above, more circumspection would be appropriate here, given the limited number of systems where this has been explicitly explored, and lingering uncertainties for even those systems

which have been explored; replace “the most drastic changes” with “some of the most drastic changes.”

We have updated the sentence.

l. 133: Hyphenate “near-modern” (compound adjective preceding a noun).

We have done so

l. 333, 340-341: I suggest changing “after the Cambrian Explosion” to “in the early Cambrian” as, by the standards of some archives (e.g., the non-SSF body fossil record) the Cambrian Explosion does not even start until ca. 520 Ma.

We have changed the sentences to ‘significant ocean anoxia after 520 Ma’ and ‘ocean anoxia in the early Cambrian’

Fig. 1, 3, 5: Check that only the euxinic subset of the Boyle et al. (2014) compilation of Mo concentration data are used here, as euxinic sediments should not be compared to non-euxinic sediments (given different potentials for trace metal enrichment). Also, there have been a considerable number of Mo studies of this time interval since publication of the compilation of Boyle et al. (2014) (e.g., Jin et al., 2016, EPSL, among many others). It would be good to add these recent data.

Following convention, we prefer to plot all of the anoxic Mo data available, but taking the reviewers point, we have added a colour-code denoting the different subsets of the data. Additionally, we have updated our dataset with the data presented in Chen et al., 2015, NComms.

Most Mo studies published after 2015 do not have an exact dating for the samples they analyzed (e.g. Jin et al., 2016, EPSL; Li et al., 2017, Geology), which makes it difficult for us to include these in our compilation. These papers (and others, e.g. Dahl et al., 2017, Geochemical Perspectives Letters) also do not suggest any different trends than are displayed in our current compilation. Therefore we believe that our current, updated compilation for this timeframe does contain the most up to date data.

We thank the reviewer for the considerable review effort. We certainly believe that we now have a much stronger manuscript that better matches the complex spatial and temporal variability that is current inherent in the available data.